# Finely tuned eye movements enhance visual acuity

Janis Intoy [1,2,3] & Michele Rucci[2,3]*

High visual acuity is essential for many tasks, from recognizing distant friends to driving a car. While much is known about how the eye's optics and anatomy contribute to spatial resolution, possible influences from eye movements are rarely considered. Yet humans incessantly move their eyes, and it has long been suggested that oculomotor activity enhances fine pattern vision. Here we examine the role of eye movements in the most common assessment of visual acuity, the Snellen eye chart. By precisely localizing gaze and actively controlling retinal stimulation, we show that fixational behavior improves acuity by more than 0.15 logMAR, at least 2 lines of the Snellen chart. This improvement is achieved by adapting both microsaccades and ocular drifts to precisely position the image on the retina and adjust its motion. These findings show that humans finely tune their fixational eye movements so that they greatly contribute to normal visual acuity.

[1] Graduate Program for Neuroscience, Boston University, Boston, MA 02215, USA. [2] Department of Brain and Cognitive Sciences, University of Rochester, Rochester, NY 14627, USA. [3] Center for Visual Science, University of Rochester, Rochester, NY 14627, USA. *email: mrucci@ur.rochester.edu

Humans critically rely on high visual acuity. Although fine spatial resolution is restricted to the foveola, a tiny region of the retina that covers less than 0.1% of the visual field, its loss has devastating consequences in everyday life, as experienced by subjects affected by deficits in foveal vision[1].

Individuals with normal or corrected-to-normal vision typically achieve a resolution of 0 logMAR—the 20/20 line of a Snellen chart—or better, which corresponds to the capability of resolving lines as thin as 1 min of arc, an astounding 1/60th of a degree (Fig. 1a). Since this level of resolution roughly matches the filtering of the optics[2] and the spacing of receptors within the foveola[3,4], it is often assumed that visual acuity is primarily determined by spatial factors, i.e., the spatial rules determining how an image can be discretized with minimal loss of information by a lattice of receptors[5–7]. However, the eyes are never stationary, and neurons in the visual system are strongly selective not just for spatial patterns, but also for temporally changing stimuli. Thus, unlike a stationary camera, it is doubtful that human acuity relies on purely spatial mechanisms.

Humans always move their eyes during the acquisition of visual information, even when attempting to maintain steady gaze on a single point (Fig. 1b). Rapid gaze shifts known as saccades typically occur 2–3 times per second, bringing a new portion of the visual scene into the foveola. In between these movements, the so-called periods of "fixation", the eyes wander incessantly, following seemingly random trajectories (ocular/eye drift) occasionally interrupted by miniature replicas of saccades (microsaccades)[8–11]. These fixational eye movements continually modulate the luminance flow impinging onto the retina and downstream neuronal activity[12–17]. Given the slow temporal integration of retinal neurons[18,19], it has long been questioned how this motion does not impair spatial resolution, resulting in a percept similar to a blurred photograph acquired by a shaky camera[20,21].

Although less known, the opposite argument has also been made. It has long been argued that eye movements could be beneficial, rather than detrimental, to visual acuity[22–25]. With stimuli far from the limits of spatial resolution, both eye drifts[26,27] and microsaccades[28] have been found to facilitate pattern vision. However, experimental evidence on acuity has been contradictory. While no study has examined the consequences of eye movements in the standard Snellen test, conflicting results have been reported in other tasks. Pioneering experiments reported no benefits from eye movements on the minimum width of bars and vernier offsets that can be detected[29,30]. In contrast, retinal image motion seems advantageous when high-acuity stimuli are directly flickered on the retina, effectively bypassing the optics of the eye[31]. The reasons for these discrepancies, whether technical limitations in the earlier studies or the differences in stimulus delivery, are presently unclear.

The recent findings with direct retinal stimulation[31] are particularly interesting because the known characteristics of fixational eye movements do not appear suited to enhance acuity. Within the sensitivity range of parvocellular (P) ganglion cells[18,19]—the neurons primarily responsible for high-acuity vision[32,33]—the power delivered by drift modulations peaks at much lower spatial frequencies than those needed for visual acuity (e.g., Fig. 2d), effectively yielding noisy signals in this range. Furthermore, very fine control of microsaccades also seems necessary to properly position the stimulus on the foveola[28]. Precisely directed microsaccades down to amplitudes of 20′ have been previously observed[28,34]. However, the separation between adjacent optotypes in a 20/20 line of the Snellen eye chart is only 10′ (Fig. 1b), and it is unclear whether microsaccades this small can be precisely directed.

Theoretical considerations[26,35] indicate that changes in drift characteristics, specifically a slower and more curved drift, would have the desired consequence of shifting power to a higher range of spatial frequencies. Do humans adjust their eye drifts to reach their acuity limits? Do they precisely direct the smallest microsaccades? And if so, how much do eye movements contribute to standard assessments of visual acuity? Until recently, investigation of these questions was prevented by the technical challenges inherent in precisely measuring eye movements, estimating how they affect the spatiotemporal input, and controlling the luminance flow on the retina. These challenges can now be overcome by means of recently developed methods for gaze-contingent control[36].

Building upon these recent advances, here we investigate the functions of eye movements in the most common test of visual acuity, the Snellen eye chart. We show that humans actively tune both major components of fixational eye movements, ocular drift and microsaccades, to benefit from the spatial and temporal properties of retinal processing. Acuity is impaired when eye movements can no longer exert their normal consequences on the luminance flow entering the eyes. These results suggest that fine control of eye movements plays a critical role in achieving the limits of visual resolution: high acuity is not a purely visual accomplishment but the outcome of a visuomotor process that requires active control.

## Results

**Tuning fixational eye movements.** To investigate the importance of fixational eye movements in visual acuity, we first examined

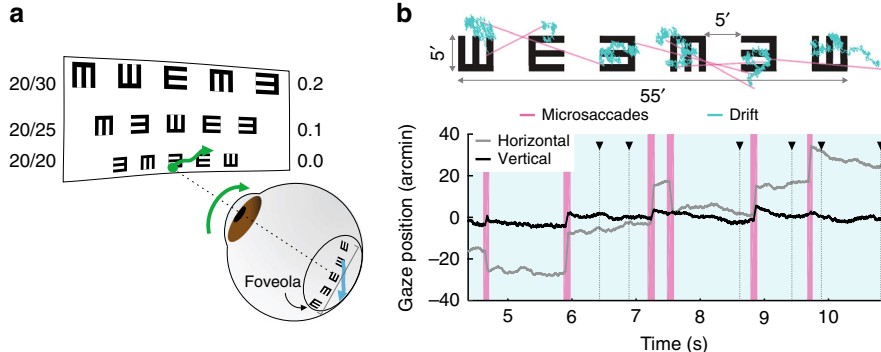

**Fig. 1 Fixational eye movements and Snellen acuity. a** Several lines from a standard eye chart. The 20/20 line corresponds to a minimum angle of resolution (MAR) of 1 arcmin (logMAR = 0). Fixational eye movements (green arrows) cause the image to move on the retina (blue arrow). **b** Example of eye movements during examination of the 20/20 line. An oculomotor trace is shown superimposed onto the stimulus (top) and over time (bottom). Green and pink colors mark the periods of drifts and microsaccades, respectively. The black triangles mark the time at which the subject reported each optotype in the array. Source data are provided as a source data file.

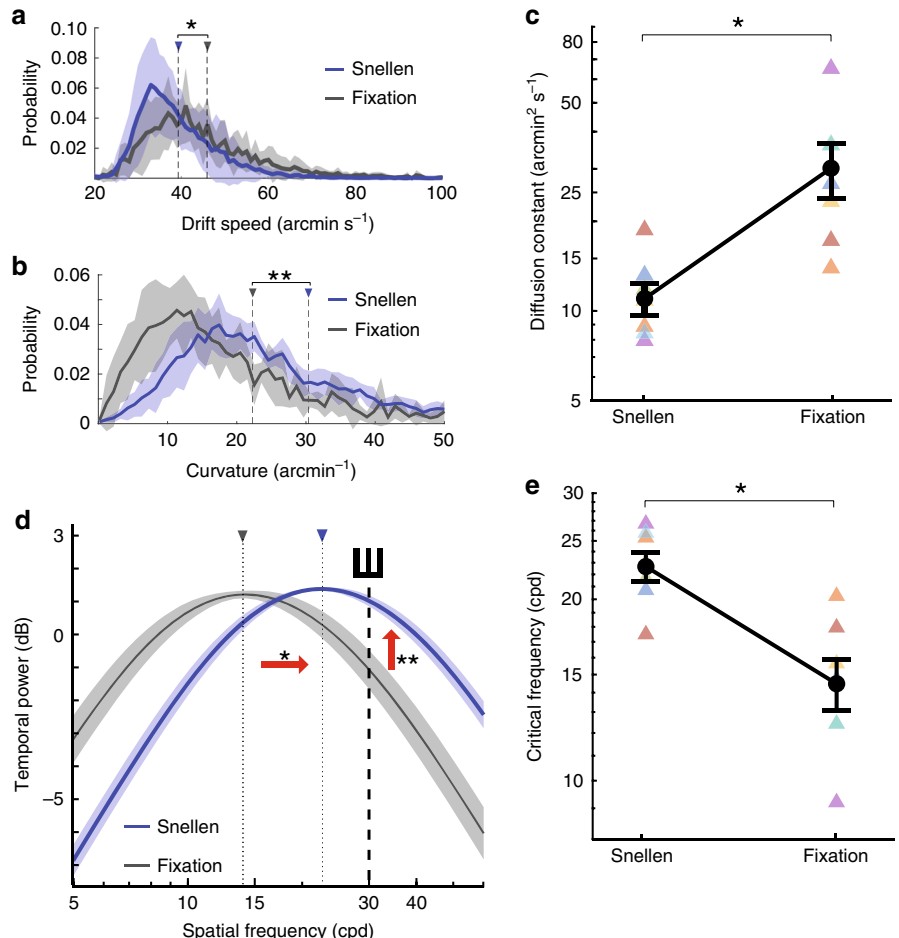

**Fig. 2 Drift characteristics.** Comparison of oculomotor data collected in the Snellen test and during sustained fixation. Data represent averages and SEM across $N = 7$ subjects. **a, b** Mean distributions of **a** drift speed and **b** curvature in the two conditions. Dashed lines indicate the means of the distributions (*$p = 0.047$, **$p = 0.031$, two-tailed Wilcoxon signed-rank test). **c** Average constants of drift diffusion, $D$, in the two conditions (*$p = 0.035$, two-tailed paired $t$-test). **d** Average power in the luminance modulations resulting from ocular drift. Power at different temporal frequencies was weighted by the temporal sensitivity of P cells and integrated to estimate the driving input for these neurons. Changes in eye drift shift modulations to higher spatial frequencies (horizontal arrow) amplifying power in the range of the optotypes (dashed line and vertical arrow). Dotted lines with triangles mark the peaks of the distributions (*$p = 0.009$, two-tailed paired $t$-test, **$p = 0.022$, two-tailed paired $t$-test). **e** Changes in the critical spatial frequency, $k_c$, the frequency that delivers the largest luminance modulations (* as in **d**). Shaded regions and error bars represent SEM. Triangles in **c** and **e** represent data from individual subjects. Source data are provided as a source data file.

whether human observers tune them to the task. To this end, we compared the characteristics of the eye movements recorded during inspection of the 20/20 line of a Snellen chart (Fig. 1, Supplementary Movie 1) with those recorded during the initial period of each trial before the appearance of the optotypes, when observers were simply asked to maintain steady gaze on a fixation marker (a 2′ dot). Sustained fixation is a standard condition for studying fixational control, and the eye movements it elicits have been extensively characterized[8–11,37].

Both ocular drift and microsaccades, the two main components of fixational eye movements, differed in important ways in the two periods. In contrast to the widespread assumption that eye drift is caused by limits in oculomotor control[38,39], striking differences in the characteristics of this movement occurred. First, drift was slower in the Snellen test, with an average speed reduction of approximately 15% relative to fixation (Fig. 2a). Second, drift was also more curved in the high-acuity task than during fixation (Fig. 2b). These differences were not just present in the average data across subjects; they were also clearly visible in the distributions exhibited by every individual observer. For each subject, the distribution of drift speed was significantly narrower

in the Snellen test compared to fixation, whereas the distribution of drift curvature was broader ($p < 0.001$, two-tailed two-sample Kolmogorov–Smirnov test).

Although, taken individually, these differences in speed and curvature may appear small, their joint action on the motion of the stimulus on the retina are profound. These two effects compound in maintaining the line of sight closer to its post-saccadic location, and their consequences can be summarized by a single parameter, the diffusion constant, $D$, in a Brownian motion model of ocular drift. This model captures many characteristics of ocular drift[35,40,41], and indeed in both the Snellen task and sustained fixation, the variance of drift displacement increased approximately linearly with time—a signature of Brownian motion ($r^2 = 0.98$ and 0.84 for Snellen and fixation, respectively). However, because of both the reduction in speed and the increment in curvature, the estimated $D$ measured in the Snellen test was much smaller than that obtained during the fixation period (Fig. 2c).

In fact, the average diffusion constant observed across our subjects during execution of the Snellen test was also substantially smaller than the corresponding values measured in two separate

groups of subjects that either maintained fixation for the entire duration of a trial, or freely observed natural scenes. In the sustained fixation group ($N = 29$), the mean diffusion constant ± SEM across observers was $17.5 ± 2.2$ arcmin$^2$ s$^{-1}$, a 50% increase ($p = 0.019$, ANOVA with post-hoc Tukey–Kramer test). In the free-viewing group ($N = 17$), $D = 26.2 ± 2.6$ arcmin$^2$ s$^{-1}$, a two-fold increase ($p < 0.001$). Thus, drift displaced the image on the retina more slowly and by a smaller amount during execution of the Snellen test, so that retinal receptors received input from narrower regions of the visual field in this task.

The observed changes in ocular drift have important repercussions on the visual flow impinging onto the retina. Previous studies have shown that ocular drift reformats spatial patterns into highly structured luminance modulations on the retina[26,35]. Specifically, at every non-zero temporal frequency the power of the resulting input signal depends non-monotonically on the spatial frequency ($k$) of the stimulus: it increases with $k$ up to a critical frequency, $k_c$, and then decreases for $k > k_c$ (Fig. 2d). Critically, $k_c$ depends on the amount of retinal image motion: it shifts toward higher spatial frequencies as the diffusion constant of motion decreases.

Because of the observed differences in drift motion, the luminance modulations delivered by drift to retinal receptors also differed in the two periods (Fig. 2d, e). Within the temporal range of sensitivity of parvocellular (P) ganglion cells[18,19], drift modulations in the Snellen task possessed considerably more power at high spatial frequencies. Whereas at fixation the average critical frequency $k_c$ lies around $15 ± 1$ cycles per degree (cpd; gray curve in Fig. 2d), the smaller $D$ measured in the Snellen test shifts $k_c$ up to approximately $23 ± 1$ cpd (blue curve in Fig. 2d).

This effect creates a high frequency range (above ~17 cpd) in which ocular drift delivers more power in the Snellen task. The optotypes in the 20/20 line are well within this amplification range, as they contain primarily energy at 30 cpd, so that the power of the input luminance flow on the retina increased by ~50% within the range of P cell sensitivity (vertical arrow in Fig. 2d). This signal was significantly stronger than the input that would have resulted from the drifts measured in other tasks. Relative to the separate group of subjects who maintained strict fixation for the entire duration of a trial, the input power in the Snellen test increased by 22% ($p = 0.002$, ANOVA with post-hoc Tukey–Kramer comparison). A 48% increment occurred relative to the eye drift recorded during free viewing of natural scenes ($p < 10^{-7}$).

Thus, by varying the amount of eye drift, the subjects in our experiments effectively amplified luminance modulations in the range of spatial frequencies relevant to the task. This signal enhancement is immediately obvious in reconstructions of the drift-induced temporal modulations impinging onto the retina (Supplementary Movie 2): a narrower and slower drift, like the one measured in the Snellen task, significantly sharpens the important edges of the optotypes in the 20/20 line of a Snellen chart.

These results were highly robust relative to the specific methods of data collection and analysis. Similar results were obtained when the Snellen oculomotor data were compared to those measured in a separate control experiment in which subjects were specifically instructed to maintain very accurate fixation for the entire duration of a trial (Supplementary Fig. 1), rather than in the period preceding the Snellen task. Additionally, the diffusion constants measured in the Snellen test were smaller than those measured in the same observers when performing a non-acuity task with the same 20/20 line (judging a ±4° tilt in the overall line; Supplementary Fig. 2).

Furthermore, conclusions were not influenced by the different durations of drift segments, which—because of the difference in microsaccade rates—were longer in the Snellen task. Indeed, very similar results were obtained by selectively focusing only on the first (or the last) 300 ms of each drift segment to make the durations of the periods of analysis identical in the two conditions. Similarly, differences in the amplitudes of the preceding saccades—on average smaller in the Snellen task (Fig. 3a)—were also inconsequential: results remained virtually identical when data analysis was restricted to drift segments preceded by microsaccades with comparable amplitudes in the two conditions. Thus, the changes in drift characteristics and their luminance modulations were robust effects, which did not depend on the specific design of our experiments.

Microsaccades were also tuned to the task. In keeping with the lower microsaccade rates generally observed in high-acuity tasks[34,42,43], microsaccades were less frequent during the Snellen test than during sustained fixation (mean rate ± SEM across observers in Snellen: $1.2 ± 0.1$ microsaccades s$^{-1}$ vs. Fixation: $2.5 ± 0.3$ microsaccades s$^{-1}$; $p = 0.016$, two-tailed Wilcoxon signed-rank test). Interestingly, the frequency of occurrence was not the only dimension in which microsaccades differed between the two tasks.

Microsaccades in the Snellen test were also much smaller and more directionally selective than when the same subjects maintained fixation on a dot. Their amplitudes were approximately half the value measured during fixation (Fig. 3a), with the 90th percentile of the distribution decreasing from approximately 40′ to 20′ between the two conditions (Fig. 3b). Notably, in the Snellen test, the distribution peaked at just 10′, an amplitude that matches the center-to-center spacing between neighboring optotypes in the 20/20 line. Furthermore, microsaccades in the Snellen test exhibited a strong bias for shifting gaze to the right, an effect reflected in their narrower angular variance relative to sustained fixation (Snellen: $0.51 ± 0.05$ vs. Fixation: $0.64 ± 0.05$; $p = 0.047$, two-tailed Wilcoxon signed-rank test; Fig. 3c). Similar effects were also observed when comparing the average microsaccade characteristics recorded in the Snellen test to those measured in the two separate groups of subjects that either maintained fixation for the entire duration of the trial or freely observed natural scenes. In both cases, microsaccades were larger than in the Snellen test (90th percentile amplitude in fixation: $33.6 ± 1.6$ arcmin; $p = 0.0013$; free-viewing: $52.7 ± 1.4$ arcmin; $p < 10^{-9}$; ANOVA with post-hoc Tukey–Kramer tests), and they were also less likely to shift gaze to the right (percentages of rightwards microsaccades in fixation: 19.6%, $p = 0.00003$; free-viewing: 15.7%, $p < 10^{-7}$).

Together, these changes in amplitude and direction shifted the average position of gaze progressively rightwards during the course of a each trial, bringing the retinal projection of each optotype close to the preferred retinal locus of fixation (PRL), the narrow region on the retina at the very center of gaze (Fig. 3d, Supplementary Movie 3). Microsaccades were highly efficient in positioning stimuli on the retina. On average, each optotype fell within 7′ from the center of gaze (Fig. 3e). Furthermore, when the microsaccades executed in the Snellen test were randomly replaced by those acquired during fixation, significant increments occurred in both (a) the average distance of each microsaccade landing to the closest optotype (Fig. 3e) and (b) the average distance of each optotype to the nearest microsaccade landing (a ~30% increase; $p = 0.0043$, two-tailed paired $t$-test). As shown in Fig. 3e, the distance by which microsaccades brought the line of sight close to an optotype also increased considerably when microsaccades were substituted by other randomly selected microsaccades occurring in the Snellen task.

Microsaccades with amplitudes close to 10′ played an important role in these effects. Both distances mentioned above remained virtually unchanged when only microsaccades in the

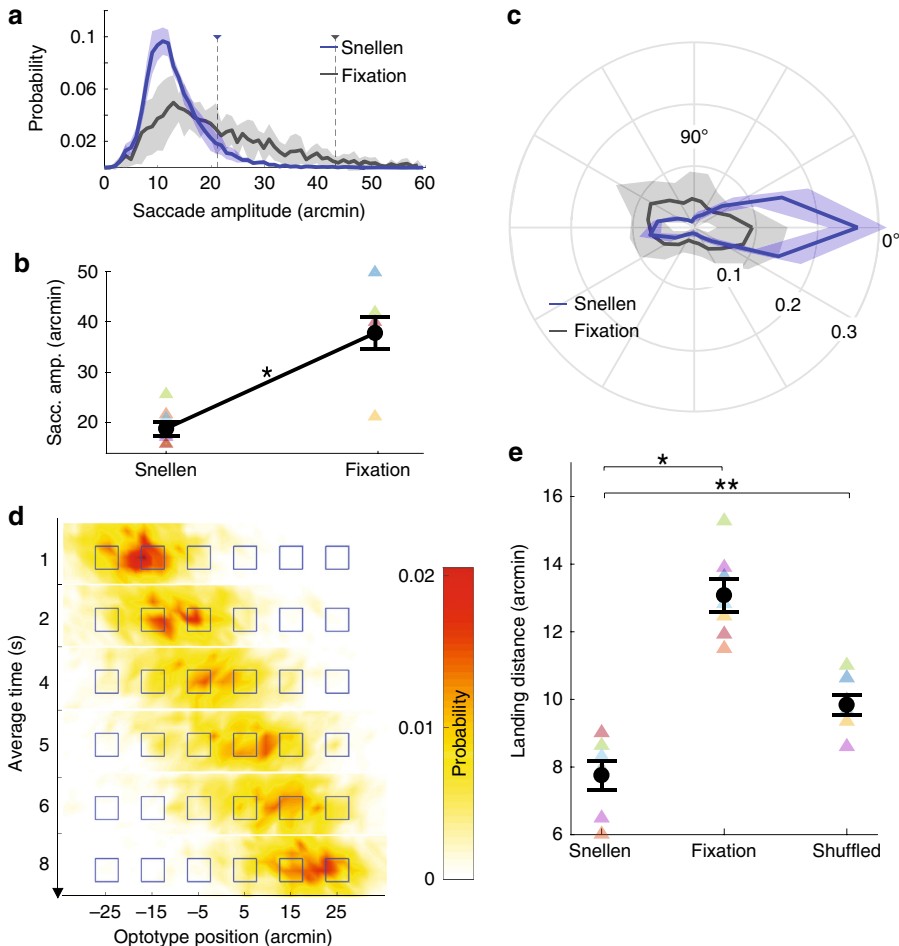

**Fig. 3 Microsaccade characteristics. a** Average distribution of microsaccade amplitudes. Oculomotor data collected in the Snellen test (blue) are compared to those measured during sustained fixation (gray). Dashed lines indicate the means of the distributions. **b** 90th percentiles of the amplitude distributions in the two conditions (*$p = 0.0009$; two-tailed paired $t$-test). **c** Average distributions of microsaccade directions. **d** Distributions of gaze probability at different times during the course of the trial. The squares mark the optotypes positions. Note the left-to-right gaze shift mediated by microsaccades. **e** Average distance between microsaccade landing and the nearest optotype. Data from the Snellen test are compared to those obtained when microsaccades in Snellen were randomly replaced by those executed during fixation (Fixation) or when they were randomly permutated (Shuffled; *$p = 0.00008$, **$p = 0.0009$; two-tailed paired $t$-test). Shaded regions and error bars represent SEM. Triangles in **b** and **e** represent data from individual subjects. Source data are provided as a source data file.

7.5′–12.5′ amplitude range were included in the analyses. When these microsaccades were randomly replaced with similar amplitude microsaccades executed during fixation, the distance from each microsaccade landing to the nearest optotype increased by 27% ($p = 0.0006$; two-tailed paired $t$-test). Also the distance between each optotype and nearest saccade landing increased significantly (by 12%; $p = 0.02$).

Like for ocular drift, these effects were very robust. Significant differences in both the microsaccade amplitude and direction distributions measured in the two tasks were present in the data from every individual observer (amplitude: $p < 10^{-9}$, two-tailed Mann–Whitney $U$-test; direction: $p \leq 0.001$, two-tailed two-sample Kuiper's test). Furthermore, the microsaccades measured in the Snellen task were significantly smaller than those measured in the same subjects when they were asked—in a separate control experiment—to maintain strict fixation for the entire duration of the trial, rather than in the period preceding the Snellen test (Supplementary Fig. 1). Microsaccades were also smaller than those measured in the same observers when performing a non-acuity task with the 20/20 line (judging a ±4° line tilt; Supplementary Fig. 2).

These results reveal an oculomotor strategy for positioning the PRL close to each individual target. This strategy is remarkable both because of the small size of the eye movements involved and the high degree of control that it entails. Precisely directed microsaccades have been previously reported in the literature[28,34]. However, the targeted movements observed by these previous studies were considerably larger than those measured here, approximately the double in amplitude. Thus, even the smallest microsaccades appear to be tailored to the needs of the task.

**Oculomotor contributions to visual acuity.** Having established that fixational eye movements, both microsaccades and ocular drift, are tuned to the Snellen test, we quantified their contributions to visual acuity. To this end, we counteracted their normal consequences on the visual flow by maintaining the stimulus immobile on the retina, a process known as retinal stabilization (Fig. 4a). This was achieved by means of a custom system for gaze-contingent control[44], which enabled real-time updating of the stimulus on the display according to the observer's eye movements. This system has been extensively tested and has been shown to yield high quality of retinal stabilization[45].

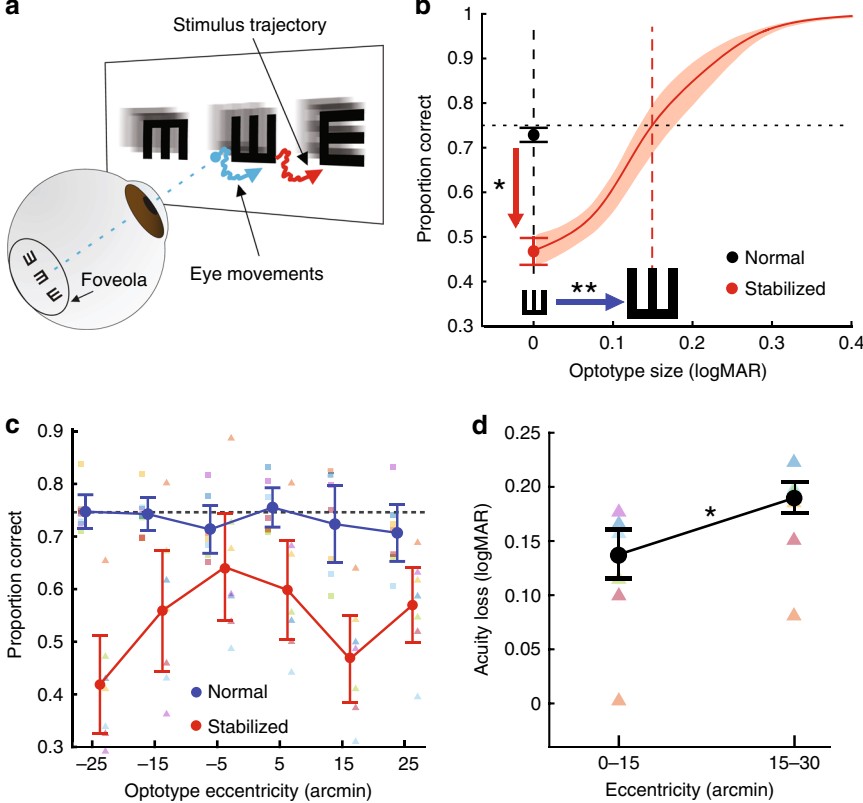

**Fig. 4 Oculomotor contributions to acuity. a** Retinal stabilization. Stimuli moved on the display under real-time computer control (red arrow) to counteract the motion of the stimulus on the retina caused by eye movements (blue arrow). **b** Performance as a function of optotype size under retinal stabilization. The red line is the average psychometric function of $N = 7$ observers. Contrast was individually adjusted to yield ~5% correct identification during normal unstabilized viewing of 0 logMAR optotypes (black circle). Retinal stabilization greatly impaired performance (red arrow). Increasing the optotype size reestablished threshold level (blue arrow). The shaded region represents SEM (*$p = 0.003$, two-tailed paired $t$-test; **$p = 0.002$, two-tailed paired $t$-test). **c** Performance by optotype position in the 20/20 line. Filled circles represent averages across subjects for optotypes at different eccentricities on the display. Performance under retinal stabilization was significantly lower at all positions ($p < 0.03$; one-tailed paired $t$-test). Squares and triangles represent individual data under normal and stabilized viewing, respectively. Error bars represent SEM. **d** Acuity loss in the central 0–15′ and in the 15′–30′ range of foveal eccentricity. Black circles and error bar represent medians and SEM across subjects (*$p = 0.0008$, two-tailed paired $t$-test). Triangles are individual subject data. Source data are provided as a source data file.

We compared performance in the discrimination of 0 logMAR optotypes (the 20/20 line of the eye chart) between two conditions: normal viewing and retinal stabilization. In the former condition, when eye movements normally moved the stimulus on the retina, the contrast of the stimulus was individually adjusted so that the mean percentage of correct identification was close to a threshold of 75% (dotted line in Fig. 4b). Performance was drastically impaired under retinal stabilization (red arrow in Fig. 4b). On average across subjects, the proportion of correct responses fell by approximately 25% when eye movements could no longer exert their visual consequences on the retina ($p = 0.003$; two-tailed paired $t$-test).

Increasing the size of the optotypes improved performance under retinal stabilization and enabled reestablishing the same threshold level obtained in the normal, unstabilized condition (blue arrow in Fig. 4b). On average, this happened by enlarging each optotype by approximately $0.15 \pm 0.02$ logMAR, a loss in the minimum angle of resolution of about 40% ($p = 0.002$; two-tailed paired $t$-test). These effects were highly consistent across subjects and statistically significant in each individual observer ($p < 0.01$; one-sample two-tailed permutation test). They correspond to a visual loss of approximately two lines of the Snellen chart: from the 20/20 line in the presence of normal retinal image motion to the 20/30 line in its absence.

The negative consequences of retinal stabilization could not be counteracted by increasing contrast. Unlike the immediate benefit resulting from enlarging the optotypes, performance under retinal stabilization improved little when contrast was increased while maintaining fixed-size optotypes. Performance remained far below threshold even at maximum contrast ($p = 0.003$; two-tailed paired $t$-test; Supplementary Fig. 3a). This marginal influence of the contrast of a stabilized stimulus is consistent with dynamic theories of spatial vision[22–26,35,46] and deviates sharply from the strong facilitation that contrast exerts during normal, unstabilized viewing (Supplementary Fig. 3b).

The adverse consequences of retinal stabilization on acuity were evident across the entire foveola, but varied with eccentricity (Fig. 4c). Around the very center of gaze (the two locations at ±5′ eccentricity), retinal stabilization resulted in an impairment in performance of approximately 16% ($p < 0.03$; one-tailed paired $t$-test). Given that these small eccentricities do not normally trigger microsaccades[28,47], this loss was presumably caused by the absence of drift luminance modulations. At eccentricities larger than 5′, retinal stabilization was even more disruptive ($p < 0.02$). This greater impairment was likely caused by the impossibility of using microsaccades to recenter the preferred retinal locus onto the more eccentric optotypes, as it occurred in the normal condition. In keeping with this idea, performance was also

impaired when image motion was normally allowed on the retina, but subjects were asked to suppress their microsaccade scanning strategy by maintaining fixation at the center of the array (Supplementary Fig. 4). Asymmetries were also visible between the nasal and temporal retina, with the majority of subjects performing better at the largest nasal eccentricity, relative, for example, to the adjacent one (the optotype at ~5′ eccentricity), an effect statistically significant in two observers ($p < 0.05$; Z-test).

Because of the dependence on eccentricity, the amount by which the stimulus had to be enlarged to recover threshold performance also varied across the foveola. The median loss in acuity increased from $0.14 \pm 0.02$ logMAR close to the preferred retinal locus (eccentricity<15′) to $0.19 \pm 0.01$ logMAR in the larger range of tested eccentricities (15′–30′; Fig. 4d). These results show how rapidly acuity drops with eccentricity across the foveola and highlight the importance of both drift and microsaccades in ensuring high acuity during normal examination of fine details.

## Discussion

Little attention is usually paid to eye movements when measuring visual acuity. Yet, the human eyes are always in motion, even when the retinal projection of the attended stimulus already falls within the high-acuity foveola, as during examination of the finest rows in an eye chart. In this study, we coupled high-resolution eye-tracking with real-time control of retinal stimulation to investigate the roles of eye movements in a standard acuity test. Our results show that humans finely tune their eye movements to enact an oculomotor strategy that takes advantage of both the spatial and temporal selectivities of retinal neurons. Visual acuity is impaired when this oculomotor strategy is prevented from exerting its normal consequences on the visual input.

This study provides important contributions on two fronts: the control of eye movements, and their visual functions. In terms of oculomotor control, our results show that humans are capable of adapting their fixational eye movements to an unexpected degree, a behavior that paradoxically leads to a more stable retinal image during an acuity test than when observers are actually asked not to move at all, as when fixating on a point. The most suprising element of this oculomotor strategy is the tuning of ocular drift. The incessant jitter of the eye is widely believed to be an involuntary, random motion, presumably resulting from physiological limits in oculomotor precision[38,39]. Contrary to this idea, it has long been suggested that eye drift may actually represent a form of controlled motion[48,49], a proposal consistent with findings of control at very low speeds in other types of eye movements, such as pursuit[50] and the vestibulo-ocular reflex[51]. Our study shows, for the first time, that humans tune their eye drift in a way that is consistent with active theories of vision[26,46]. The changes in drift speed and curvature measured during examination of the finest lines of a Snellen chart are functionally important, as they increase power within the frequency range relevant to the task (Fig. 2d). This effect can be directly observed in reconstructions of the spatiotemporal flow impinging onto the retina (see Supplementary Movie 2).

Microsaccades also exhibited a remarkable degree of control. They redirected the line of sight from one optotype to the next, even though the optotypes were only 10′ apart. Control of microsaccades has been previously reported in the literature[28,34,40]. In particular, using methods for accurate gaze localization similar to the ones employed here, two previous studies have observed targeted microsaccades[28,34]. Critically, however, microsaccades in these previous studies were considerably larger, almost the double of those measured here. Microsaccades with amplitudes around 10′ were rare[34] or virtually absent[28], and

determination of whether such small gaze shifts were also targeted was not possible.

For example, in a simulated needle-threading task, microsaccades shifted the line of sight back and forth between the tip of the thread and the eye of the needle[34]; these movements became smaller as the thread approached the needle, but their average amplitude at the end of the task was still close to 20′, which matched the microsaccade amplitude measured from the same observers during sustained fixation. In the Snellen test, the peak amplitude of microsaccades was only 10′ and matched the spacing between adjacent optotypes. These movements were precisely directed, as revealed by their landing positions and by their random permutations or substitutions. Thus, during normal examination of stimuli at the limits of visual resolution, precise control extends to very small microsaccade amplitudes.

In terms of visual functions, by showing that eye movements play a fundamental role in the outcome of a standard acuity test, this study provides support to the so-called dynamic theories of visual acuity[22–26], the long-standing idea that oculomotor activity is instrumental for acuity. Reduced performance in the absence of retinal image motion has been recently reported with stimuli directly projected on the retina[31]. In this previous study, isolated optotypes at the limits of resolution were flickered at 30 Hz—a low frequency that affects contrast sensitivity[52]—while counteracting for the effects of the eye optics. However, earlier studies with more natural stimulation reached the opposite conclusion of no effects of eye movements on acuity[29,30], and the reasons for this discrepancy have remained unclear.

Our data show that, contrary to these older reports, the impairments observed with direct retinal stimulation extend to more natural conditions, with stimuli displayed at high refresh rates and normally viewed through the eye optics. Critically, our results go beyond the previous literature in several important ways, including (a) the finding of oculomotor tuning in both drifts and microsaccades; (b) the observations that these motor behaviors increase power in the range of neuronal sensitivity and enable precise re-centering of the stimulus on the retina; (c) the estimation of the acuity loss resulting from lack of visuomotor consequences, i.e., how much optotypes need to be enlarged to maintain performance; and (d) the quantification of the consequences of eye movements in the Snellen eye chart, the most common test of visual acuity, where stimuli are not isolated and their layout plays an important role.

Regarding the mechanisms by which eye movements enhance acuity, several possibilities remain open. One possibility is via spatial mechanisms similar to the super-resolution algorithms developed in computer vision[53,54]. These algorithms enable estimation of higher resolution images than those afforded by the sensor in the camera. A similar approach, in which the motion of the image enables overcoming sampling limitations imposed by the receptor array in the retina, was favored by Ratnam et al. (2017)[31] as an explanation for their findings. An alternative explanation, in principle not mutually exclusive with the previous one, relies on the characteristics of the spatiotemporal flow impinging onto the retina. With larger stimuli—stimuli far from the limits of acuity—beneficial influences from fixational eye movements have been previously reported: both the temporal luminance modulations resulting from eye drifts[26,35] and the positioning of the stimulus on the retina operated by microsaccades[28] enhance foveal vision. Without the oculomotor adaptation observed in our experiments, these effects would not extend to visual acuity, as the characteristics of both microsaccades and drifts typically measured in non-acuity tasks are too coarse to enhance features at this scale: drift modulations would peak at too low spatial frequencies, and re-centering of each optotype on the preferred retinal locus would be difficult without

control of the smallest microsaccades. However, the observed oculomotor tuning extends these benefits to to the limits of spatial resolution, suggesting that the strive for acuity is the primary factor driving these behaviors. While other factors unrelated to acuity could also contribute to the production of these visuomotor strategies, acuity is impaired in their absence, as during retinal stabilization or when examining a Snellen line in the absence of microsaccades.

Our findings suggest a possible link between deficits in visual acuity and fixational eye movements. Oculomotor activity is rarely monitored during assessment of visual acuity, and poor outcomes in the Snellen test are commonly attributed to defects in the optical, structural, and/or physiological properties of the eye, not eye movements. Yet, abnormal fixational eye movements and impairments in fine spatial vision co-occur in multiple disorders. For example, poor fixational control accompanies the visual impairments present in conditions such as amblyopia[55] and dyslexia[56], and reduced visual acuity co-exists with the motor abnormalities present in conditions such as nystagmus[57,58] and Parkinson's disease[59,60]. That is, all these conditions exhibit both abnormal fixational eye movements and impaired acuity. Furthermore, theoretical considerations suggest that chronic exposure to the retinal input resulting from poor fixational control affects the maturation of the receptive fields of cortical neurons during development[61–63]. These considerations together with the findings of our study point at the need to examine in greater depth the consequences of abnormal eye movements for visual acuity.

The acuity impairment measured in our experiments likely underestimates the real contribution of eye movements. One reason for this has to do with the way the Snellen chart itself is structured. While this eye chart represents the most widespread method for measuring visual acuity, it does not control for the possible effects of crowding, the negative consequences on visibility exerted by nearby stimuli[64]. In the Snellen chart, the number of optotypes does not remain constant across rows, but increases as the optotypes become smaller. This implies that the increased difficulty of the task with finer optotypes may stem not just from the required higher acuity, but also from more severe crowding. In our experiments, this effect may have partly compensated for the impairment caused by stabilization when the optotypes were progressively enlarged.

Furthermore, one has to keep in mind that perfect retinal stabilization is not experimentally achievable, and theoretical considerations suggest that stabilization errors may also contribute to underestimating the real impact of eye movements. Our apparatus provides state-of-the-art quality of retinal stabilization, leaving a residual motion on the retina of approximately 1′[45]. The resulting luminance modulations are not just smaller in amplitude, they also emphasize higher spatial frequencies. This happens because reducing the scale of retinal image motion is functionally equivalent to enlarging the stimulus, which translates into a compression of the spatial frequency axis in the frequency domain. As a consequence, while the strength of the input flow is reduced under retinal stabilization, it may still provide useful temporal power in the spatial frequency range of the smallest optotypes. Changes in the shape of the contrast sensitivity function under retinal stabilization are consistent with the idea that the visual system is sensitive to this residual motion[65].

It is worth pointing out that our results appear to have little to do with image fading, the gradual disappearance of stimuli observed under prolonged retinal stabilization. Image fading is typically observed with low-contrast and low spatial frequency stimuli displayed far from the fovea[66]. In agreement with previous studies[31,45], none of our participants reported fading with the sharp, high-contrast foveal stimuli of our experiments. In fact,

under retinal stabilization, recovery of threshold performance could only be achieved by increasing the size of the optotypes, not their contrast (Supplementary Fig. 3). This behavior deviates from the strong beneficial influence exerted by contrast in the presence of the physiological motion of the retinal image. Such changes support the notion that the visual system uses luminance modulations from eye movements to encode spatial information[22–26,46].

In sum, our results show that humans fine-tune their eye movements in tasks at the limits of spatial resolution. The resulting motion of the image on the retina plays a critical role in the outcome of the most common assessment of visual acuity. These results suggest that low performance in acuity measurements may result from suboptimal eye movements and stress the importance to carefully examine fixational eye movements in subjects with impaired acuity.

## Methods

**Subjects.** A total of 13 emmetropic subjects participated in the main experiments of this study (6 females and 7 males; average age: 23; age range: 20–35): seven subjects took part in the experiments of Figs. 2–4 and Supplementary Figs. 3 and 4; six other subjects participated in the control experiments of Supplementary Figs. 1 and 2. The oculomotor data collected in these experiments were compared to those collected from other 46 subjects (21 females and 25 males; average age: 22), who either maintained fixation or freely examined pictures of natural scenes. Subjects were naive about the purpose of the study and were compensated for their participation. To qualify, subjects had to possess at least 20/20 acuity in the right eye, which was assessed by correct identification of at least 75% of the optotypes in the 20/20 line during a standard execution of the Snellen test. Experiments followed the ethical procedures approved by the Charles River Campus Institutional Review Board at Boston University and the Research Subjects Review Board at the University of Rochester. Informed consent was obtained from all subjects.

**Stimuli and apparatus.** Stimuli consisted of horizontal arrays of black tumbling-E optotypes displayed over a white uniform background (14 cd m$^{-2}$). They were displayed at the center of a calibrated fast-phosphor CRT (Iiyama HM204DT; 1024 × 768 pixel resolution, 150 Hz refresh rate) placed in front of the observer in a dimly illuminated room. In every trial, a single array was presented with all the optotypes of equal size and contrast. Each optotype had equal probability to be oriented along four possible directions (legs up, down, left, or right). The size of the optotypes ranged from 5′ to 16′, with adjacent optotypes always separated by a space equal to the optotype width. Stimuli were viewed monocularly with the right eye, while the left eye was patched. A dental imprint bite-bar and head-rest minimized head movements and maintained the observer at a fixed distance from the display.

Vertical and horizontal eye position data were measured by means of a Dual Purkinje Image (DPI) eye-tracker (Fourward Technology). Analog oculomotor signals were low-pass filtered at a cutoff frequency of 500 Hz and sampled at 1 kHz. Stimuli were rendered by means of EyeRIS, a hardware/software system for gaze-contingent display control that enables precise synchronization between eye movement data and the refresh of the image on the monitor[44].

**Experimental procedures.** Data were collected in multiple experimental sessions, each lasting approximately 1 h. Every session started with preliminary steps aimed at ensuring optimal eye-tracking and gaze-contingent control. Blocks of trials then followed, each lasting 10–15 min. Breaks in between blocks allowed the subject to rest.

To achieve localization of the line of sight, subjects underwent a two-stage calibration procedure. In the first phase, they sequentially looked at each of the 9 points of a standard 3 × 3 grid. This yielded a first estimate of the parameters of a bilinear transformation that mapped DPI voltages into visual angles. Parameters were then refined in a second gaze-contingent phase, in which subjects manually fine-tuned the estimated position of gaze, displayed in real-time on the monitor for each of the grid points. This approach enables accurate determination of the intersection between the line of sight and the display—i.e., the point in the stimulus that projects onto the center of the preferred retinal locus of fixation (PRL). This method has been shown to increase the accuracy of gaze localization by approximately one order of magnitude relative to standard eye-tracking calibrations[36]. Note that this procedure estimates the distance of a stimulus from the PRL in visual field coordinates; it does not allow determination of where the PRL is located on the observer's retina. To counteract possible misalignment caused by drifts in the apparatus and/or minute head movements, the gaze-contingent procedure was repeated for the central fixation point before every trial.

In a forced-choice procedure, subjects sequentially reported the orientations of all the optotypes in the array. They were instructed to proceed from left to right, as in a standard Snellen test, using four keys on a joypad. Each trial started with the

subject maintaining strict fixation on a 2′ dot at the center of a uniform field for 1 s. The stimulus then appeared with the contrast of the array gradually increasing over the course of 1 s and then remaining at a fixed value until the subject had completed the task. To help maintain track of progress within a trial, a brief sound marked the registration of each response. The trial ended, and the stimulus disappeared, after reporting the orientation of the last optotype. At 0 logMAR (the 20/20 line), the average trial duration was 12 s.

Blocks of trials alternated between two conditions. In the normal condition, stimuli remained at a fixed location of the display and moved normally on the retina because of eye movements. In the stabilized condition, the entire array of optotypes moved with the eye, under EyeRIS control, to counteract the consequences of eye movements and minimize retinal image motion. In the normal condition, subjects were always presented with the 20/20 line, a row of six optotypes, each 5′ in width (0 logMAR). The contrast of the optotypes varied adaptively across trials, following the Parametric Estimation by Sequential Testing (PEST) procedure[67] to determine the contrast value yielding 75% correct discrimination for each individual observer.

In the stabilized condition, we examined the effect of varying the optotype size. The contrast of the array remained fixed at the individual threshold value established in the normal condition, while the optotypes were systematically enlarged to determine the angle of resolution needed to reestablish threshold performance (method of constant stimuli: 11 optotypes within 5′–16′). As in a standard Snellen chart, the number of optotypes in the array decreased as they became larger, with the entire array spanning no more than 1°.

Four control experiments examined possible influences from various factors. Two experiments focused on performance in the Snellen test with 0 logMAR optotypes. The experiment of Supplementary Fig. 3 examined the consequence of varying contrast under retinal stabilization. Procedures were identical to those of the normal condition in Fig. 4, except that stimuli were now stabilized on the retina. The experiment of Supplementary Fig. 4 examined the effect of actively suppressing the microsaccade sequence. Subjects were asked to maintain fixation for the entire duration of the trial on a 2′ dot, which was presented at the center of the array. This request overruled the normal microsaccade behavior. Optotypes were displayed at the contrast threshold level determined in the normal condition.

Two further experiments focused on the characteristics of eye movements. In the experiment of Supplementary Fig. 1, the oculomotor data collected in the Snellen test were compared to those acquired when observers maintained fixation for the entire duration of the trial (1.5 s), rather than in the initial period preceding the Snellen task. Subjects were instructed to fixate as accurately as possible on the fixation dot at the center of the display, which was presented at maximum contrast over a uniform gray background. In the control of Supplementary Fig. 2, subjects judged whether the entire 0 logMAR Snellen line was tilted by ±4° relative to the horizontal axis. The orientation of the line randomly alternated between trials. The optotypes were always presented at the contrast threshold level determined in the normal condition.

Data collected in the Snellen task were also compared to those acquired in separate experiments in which subjects either maintained fixation or freely observed natural scenes. The procedures of these experiments were similar to those described in previous publications[35,37]. In the fixation condition, subject were asked to fixate as accurately as possible on a marker at the center of the display (a 4′ dot) for at least 1.5 s. The fixation marker was displayed at maximum contrast over a uniform background, and no other task preceded or followed fixation. In the free-viewing condition, subjects were instructed to memorize grayscale pictures of natural scenes, which were were displayed sequentially, each for 10 s. Each pixel subtended 1′, an angle similar to that covered when the image was originally acquired.

**Data analysis.** All effects are reported as group statistics. Individual statistics are reported for the most important results to point out that these are very robust effects, clearly visible in the data from each individual observer. Contrast thresholds yielding 75% correct identification were obtained by fitting a cumulative normal function to the data via a maximum likelihood procedure[68] (Supplementary Fig. 3a). A similar approach was followed to estimate visual acuity thresholds using the cumulative Weibull function (Fig. 4b). This function, used by previous studies[4,69], better fits acuity data than a cumulative normal. For each subject, we tested whether the change in the minimum angle of resolution was significant by measuring acuity thresholds via parametric bootstrap on the responses to individual optotypes ($N = 1000$).

Oculomotor data from different periods in a trial were examined separately. The first 200 ms of each trial were discarded to ensure that subjects had acquired fixation. Fixation data in Figs. 2 and 3 refer to the remaining 800 ms period of each trial, when the fixation marker was displayed. Snellen data refer to the later period in which subjects reported the optotype orientations. This period started with the visual fixation preceding the first response and ended with the reporting of the last optotype.

Oculomotor traces were segmented into complementary periods of saccades and drifts based on velocity. Events in which eye speed exceeded $3° s^{-1}$ were classified as saccades, with onset and offsets marked as the times at which the speed reached $2° s^{-1}$. Consecutive events separated by less than 15 ms were automatically merged to exclude post-saccadic overshoots. Remaining trace segments were classified as eye drifts. All events were classified automatically and verified visually. All trials with suboptimal eye-tracking or in which the subject looked away from

the stimulus (trials with saccades larger than 2°) were discarded from data analysis. Blinks were detected automatically by the DPI eye-tracker as the sudden loss and recovery of both Purkinje images. They were removed from analysis together with their surrounding segments.

Saccade amplitudes and directions (Fig. 3a–c) were determined based on the difference between eye positions at saccade onset and offset. The distributions of saccade amplitudes and directions in the Snellen task and during fixation were compared by means of the Kolomogorov–Smirnov and Kuiper's tests, respectively. The latter, being circularly invariant, is better suited for comparing angular directions. To take into account the width of the distribution, we compare microsaccade amplitudes by using their 90th percentile. The distribution of gaze position over time reported in Fig. 3d and Supplementary Movie 3 represent the average across subjects. Since no time limits were posed on the completion of the task, some trials took longer than others. To discount this variability, each trial was normalized by its duration and then subdivided into consecutive intervals (6 bins in Fig. 3d; 20 bins in Supplementary Movie 3). The time label on the y-axis of Fig. 3d indicates the average time of all the data points contained in the corresponding bin.

In Fig. 3e, data points represent the distance from the landing position of a microsaccade to the nearest optotype, averaged across all microsaccades. These measurements are compared to those obtained in two conditions: when microsaccades in the Snellen task were substituted by microsaccades randomly selected from (a) the pool recorded during fixation and (b) the pool of microsaccades recorded in the Snellen task. In both cases, the new microsaccade was positioned so to possess the same starting position as the original one. Similar analyses were conducted to also estimate the distance between the center of each optotype and the nearest saccade landing position, averaged across optotypes. To evaluate the effectiveness of 10′ saccades, these analyses were repeated considering only saccades in 7.5′–12.5′ amplitude range.

To attenuate the impact of measurement noise during the low-velocity intersaccadic periods, drift segments were filtered by means of a low-pass third-order Savitzky-Golay filter with cutoff frequency at approximately ~30 Hz. Drift periods within 50 ms from saccades were discarded from data analysis to eliminate possible saccadic influences. Results in Fig. 2 are averages across all drift segments, independent of their durations. Virtually identical results were obtained on the initial, or the final, 300 ms of each drift segment or when only drifts following microsaccades smaller than 30′ were considered.

**Spectral analysis of retinal input.** We estimated the power spectrum of the luminance modulations delivered by ocular drift on the retina. To this end, we used a Brownian motion model of ocular drift, a model that allows analytical formulation of the gain, $Q$, by which eye movements redistribute the power of the stimulus[35]:

$$Q(\mathbf{k}, \omega; D) = \frac{2D\mathbf{k}^2}{D^2\mathbf{k}^4 + \omega^2}, \tag{1}$$

where $\mathbf{k} = (k_x, k_y)$ represents spatial frequency, $\omega$ temporal frequency, and $D$ the diffusion constant of motion.

We first fitted the model for each observer, by estimating the equivalent diffusion constant of eye drift (Fig. 2c). This was accomplished by linear regression of the variance of the eye displacement as a function of time: $\sigma^2(t) \propto 4Dt$. We then measured the average power made available by eye motion at all spatial frequencies. We specifically examined the spatial frequency at which the distribution peaked (Fig. 2e) and how the change in diffusion constant in the Snellen task affected power at 30 cpd, the main frequency of a 0 logMAR optotype (Fig. 2d).

To quantify the efficacy of the visual flow in driving neural responses, spectral distributions were weighted by the temporal frequency sensitivity of parvocellular ganglion cells. This was modeled by a series of filters, as previously proposed[70]:

$$H(\omega) = A \exp(-\mathrm{i}\omega d)\left(1 - \frac{H_S}{1 + \mathrm{i}\omega\tau_S}\right)\left(\frac{1}{1 + \mathrm{i}\omega\tau_L}\right)^{N_L}. \tag{2}$$

Parameters were adjusted based on neurophysiological data[19]: $A = 12.63$, $d = 0.0022$, $H_S = 0.62$, $N_L = 46.15$, $\tau_S = 0.0259$, $\tau_L = 0.0012$. Since drift changes the spectral distribution of the input signal to the retina before any neural filter, our results are extremely robust with respect to the specific values of these parameters.

**Reporting summary.** Further information on research design is available in the Nature Research Reporting Summary linked to this article.

## Data availability

A reporting summary for this article is available as a Supplementary Information file. The source data underlying Figs. 2, 3a–c, e, and 4b–d and Supplementary Figs. 1–4 are provided as a Source Data file. All other data supporting the findings reported here are available upon reasonable request from the corresponding author.

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

## Acknowledgements

This work was supported by National Institutes of Health grants EY18363 and EY029565 and National Science Foundation grants 1457238 and 1420212. We thank Martina Poletti, Michele A. Cox, and Jonathan Victor for helpful comments and discussions throughout the course of this research.

## Author contributions

J.I. implemented the experiments, collected, and analyzed experimental data. M.R. conceived and supervised the project. Both authors contributed to the design of the experiments and the writing of the article.

## Competing interests

The authors declare no competing interests.
