## [Peer Review File · Nature Communications]

Reviewers' Comments:

Reviewer #1:

Remarks to the Author:

Visual acuity is generally known to be limited by the eye's optics and anatomy, such as the spacing of the photoreceptors on the retina. This study reveals a significant contribution of eye movements to Snellen acuity and shows that human microsaccades and drifts can enhance acuity by repositioning the image (here, a letter on a Snellen chart) on the retina. The authors convincingly show this by capitalizing on their gaze-contingent display set up with the dual-Purkinje image eye tracker, a high-resolution and precision machine. Human observer's (n=7) fixational eye movements were tuned to the task, e.g., they showed less frequent microsaccades during the Snellen test than during sustained fixation. Importantly, eye movements during the Snellen test were also smaller – with the amplitude peak matching the distance between letters – and more directionally selective (rightward shift in the reading direction); drift was slower and differed in curvature. After establishing such task tuning in fixational eye movements, the authors then quantified the effect of eye movements on acuity by comparing visual acuity during retinal stabilization and normal viewing. Acuity was reduced by approx. 25% across observers during retinal stabilization, equivalent to 2 lines on the Snellen chart.

This is a novel and incredibly important finding that will likely have impact on our scientific understanding of visual acuity as well as on clinical practice. The paper is extremely well written, and I only have a few comments on issues I felt could be addressed to further improve the paper.

I.37-40: for a reader not well-versed in eye movement vocabulary this can be a dense sentence to parse with its multiple parentheses. Consider rephrasing / splitting into two sentences to first explain macroscopic gaze shifts (saccades), and then small fixational eye movements.

I.40: I generally find the use of the abbreviation "FEM" unhelpful, perhaps because it is too close to "FEF" for my taste; it slowed down my reading rather than speeding it up

I.47: "for" these ideas

I.52-54: unclear what this sentence is really saying; in what way are these movements too coarse?

Fig 1: increase space between panels a and b, the figure looks crowded. Unclear what the significance of the vertical pink lines in panel b is – onset of microsaccades?

I.95: tuned "to" the task

I.105 following: this paragraph is a little less intuitive. Whereas the reduction of drift speed in the Snellen task makes sense, I am not sure what the significance of curvature of drift is in general. It might be helpful to explain this before outlining results (maybe moving the logic outlined in I.123-124 or I.153 following up?).

I.118: delete "well"

Fig. 4c: can the authors explain the increase in performance at the +25' eccentricity?

Fig. 4c/d: Overall, there appears to be large individual variability between subjects in terms of performance / acuity as a function of eccentricity. Were there any effects of age or other factors that might explain these differences?

Discussion: I think there is a missed opportunity here in not pointing out at least the potential relation between deficits in fixational eye movements (such as square-wave jerks, frequently found in many patients with neurological or psychiatric disorders) and acuity. Is there any indication in the literature that such a relation might exist? If not, this would certainly be an important future research direction.

I.302: how was it determined that subjects were emmetropic? Given individual differences, this is important information to add.

Reviewer #2:

Remarks to the Author:

In this elegant set of experiments, Intoy and Rucci demonstrated the role of fixational eye movements (FEM) in visual acuity, as measured by the Snellen eye chart test. They showed that the distributions of FEM amplitudes and directions change during the Snellen task compared to a fixation task, that microsaccades move precisely from one optotype to the next, that changes in drift during the Snellen task enhance high spatial frequency power, and that gaze stabilization reduces acuity by ~ 0.15 logMAR. The execution of the study is exemplary and the data are clear. I also agree with the authors that the demonstration that FEMs substantially impact acuity in the Snellen task is of potential clinical importance.

My main concern with the way the paper is presented is that I believe it may overemphasize the novelty and surprise of the findings:

1. The authors write, "It has remained unclear whether FEMs enhance acuity under natural viewing conditions" (line 224). But a recent study (Ratnam et al., 2017, cited by the authors) demonstrated impaired acuity when the image was stabilized on the retina, similar to the present study. The studies cited to the contrary were from the 1950s/60s and are clearly out of date due to technological advances since then.

2. The authors write that small microsaccades are "widely believed not to be controlled" (line 102) or "commonly believed not to be controlled" (line 246). These statements are given without references. However it is my understanding that the senior author and his colleagues have shown in several high-profile papers that microsaccades are precisely controlled. For example, Poletti, Listorti, and Rucci (2013) have shown that microsaccades can position stimuli precisely on the retina, at distances down to 5 arcmin. And Ko, Poletti, and Rucci (2010) have shown that microsaccades are tuned to a needle threading task, and closely track the virtual thread, at a fine spatial scale of apparently < 10 arcmin.

3. The authors argue, "one would not expect [the known MS enhancement of high SFs] to apply to visual acuity, as the reported characteristics of FEMs ... are too coarse" (line 229). It would be helpful to cite the papers showing coarse FEMs, and to explain what those FEM characteristics quantitatively predict for SF enhancement. But this claim also seems at odds with the previous demonstration that FEMs enhance acuity (Ratnam et al., 2017).

Questions:

1. Was the greater directional selectivity during the Snellen test due only to the microsaccades moving between optotypes (reading left to right across the row), or were microsaccades within optotypes also more directionally selective?

2. Why are some statistical tests reported for group data and some for individual observers? Please explain or report both for all tests.

3. Line 89: Why Kuiper's test instead of K-S test? (which was used in seemingly the same analysis in line 113)

4. Line 92: How was the PRL determined?

5. Figure 2c: What kind of binning was done on the "Average Time" axis, or how was the visualization

along that axis created?

6. Figure 4c: Why is the "stabilized" function asymmetrical?

7. Ratnam et al. 2017 propose a model for how FEMs enhance acuity related to the way that the cone mosaic samples the drifting image. How does this model relate to the authors' explanation? This should be discussed.

8. It may not be necessary for this report, but I wondered, what are the characteristics of eye movements during stabilized viewing? Do they differ from those during normal viewing?

Reviewer #3:

Remarks to the Author:

The key conclusions of this paper are that (1) fixational eye movements (microsaccades and drift; FEMs) contribute substantially to measurements of visual acuity in the classical clinical test (the Snellen chart), suggesting that they should be measured (or at least considered) in clinical settings, and; (2) Fixational eye movements are "tuned" to improve performance on the acuity task, including differences in the rates, amplitudes, and directions of microsaccades, and the variance of drift. The results are interpreted as evidence for the use of eye-movement-induced temporal modulations to improve sensitivity in acuity tasks.

These claims are based on two key sets of comparisons. The first is a comparison of the spatial aspects of FEMs during the Snellen task and a traditional fixation condition. The second is a comparison of performance with and without stabilization of the Snellen array on the retina (achieved via the Rucci lab's well-established techniques).

The results are interesting and their implications are broad, but it isn't clear the extent to which they represent a major advance on earlier published work. The authors give a sense of such an advance (e.g. by suggesting FEMs were previously generally considered too "coarse"), but it is left relatively vague and with un-cited assertions about the prevailing view in the field. This group has previously shown that FEMs have important effects on sensitivity to fine detail through their effects on the spatiotemporal content of incoming images. The novel advance here could thus perhaps be clarified.

Moreover, there are methodological and analysis issue that make it difficult to assess the robustness of some key results. In particular, the Snellen and Fixation conditions do not seem directly comparable. These issues are detailed below.

In the first half of the paper, FEMs are compared between the Snellen and fixation conditions. Many differences are observed, and they are interpreted as evidence for "tuned" control of FEMs due to the need for high acuity in the task. These conditions, however, are different in many important ways, not just in the need for high acuity. The comparison seems to entirely be one of apples and oranges. Some differences and observations are as follows:

a) Although not stated in the main text, the "fixation task" turns out not to be a separate, comparable task (e.g. one that differs only in the need for acuity,) but is rather the initial period of fixation in the Snellen task, before the array of optotypes is presented. At a minimum, this should be made clear early in the main text, but it could also impact on the oculomotor behaviour observed:

- What if subjects consider that epoch to be less important, as they anticipate the "real" task, and maintain less precise fixational control? (the details of how fixation was enforced were also not provided). Indeed, although the distribution of microsaccade amplitudes in the Snellen task was shifted toward smaller eye movements compared to fixation in *this* data set, it looks very similar to those seen during fixation in other studies (e.g. compare with Box 1 in "Microsaccades: a neurophysiological analysis" by Martinez-Conde et al., or "Microsaccades during reading" by Bowers and Poletti).

Similarly, stimulus and task-related asymmetries in microsaccade directions are also well established. The Rucci lab is certainly a world-leader in this area, and thus I expect that there are important advances in the result here to be appreciated, but this needs to be made clearer to the reader.

b) The Snellen array is arranged horizontally, and subjects were instructed to report the identity of the optotypes one-by-one with a joypad and from left-to-right (this important detail was also not stated until the methods). This makes sense - it's consistent with the clinical version of the task - but it is probably not surprising that in such a task the FEMs showed a left-to-right pregression. That is, again, it might have nothing to do with the need for acuity per se. In comparison, the fixation task is radially symmetric. Perhaps the same pattern of FEMs would occur with any task requiring observers to shift their average fixation position rightward over time (as was the case in the paper on reading cited above).

c) As an indication of how targeted microsaccades were, the authors state that "when the oculomotor traces recorded in the Snellen test were randomly replaced by traces of equal duration acquired during fixation, the average distance between each optotype and the PRL increased drastically (Fig. 2d)." Do they mean the average distance between the PRL and the *nearest* optotype? I presume so, because otherwise I would expect the opposite result.

d) Related to these targeted microsaccades. How does this differ from previous reports of targeted microsaccades (e.g. correcting for drift during fixation)? The suggestion is that those observed here are much smaller, but as stated above, the advance is asserted to be true without sufficient quantitative comparison to earlier findings. For example, as I have noted, the distributions of microsaccade amplitudes do not seem much smaller than in earlier work.

Minor points:

- Fig 4. The x-axis is labelled "Acuity loss", but perhaps it would be clearer to label this as "optotype size"?

- Methods state that blinks were removed, but no details provided.

- Fig 4 caption: "red circles". There is only one red circle data point.

Reply to Reviewer 1

Reviewer 1 gave a positive evaluation of the manuscript and made several helpful comments on how to further improve it. The Reviewer found the paper extremely well written and the study to provide a novel and incredibly important that will likely have impact on our scientific understanding of visual acuity as well as on clinical practice." We thank the Reviewer for her supportive comments. We have carefully modified the manuscript to incorporate all the Reviewer's suggestions, as detailed below.

Lines 37-40. For a reader not well-versed in eye movement vocabulary this can be a dense sentence to parse with its multiple parentheses. Consider rephrasing/splitting into two sentences to explain macroscopic gaze shifts (saccades), and then small eye movements.

We agree, this sentence was indeed convoluted. We have revised the text as recommended by the Reviewer. Two separate sentences now introduce eye movements at different scales.

Line 40. I generally found the use of the abbreviation FEM unhelpful, perhaps because it is too close to FEF for my taste; it slowed down my reading rather than speeding it up.

We eliminated the abbreviation as recommended by the Reviewer. In a few cases, that led to a reorganization of the text to avoid long sentences.

Line 47. "for" these ideas. We have rewritten the sentence.

Lines 52-54. Unclear what this sentence is really saying; in what way are these movements too coarse?

This is an important issue, which unfortunately was not explained clearly in the previous version of the manuscript. Our point is that the characteristics of eye movements| both eye drift and microsaccades| typically reported by studies in the do not seem well-suited to enhance visual acuity.

In terms of ocular drift, the motion normally observed during (e.g., Cherici et al., 2012) delivers luminance modulations to retinal receptors that peak at much lower spatial frequencies than those needed to report stimuli at the limits of visual acuity. An example is given in Figure 2D, which shows how the power of the input signal measured during sustained declines rapidly in the range relevant to the Snellen test (primarily 30 cycles/deg for the 20/20 line). We knew from previous studies (Kuang et al., 2012), that a reduction in the drift's diffusion constant (i.e., a slower and more curved drift) would have the consequence of shifting power to a higher range of spatial frequencies, so we were interested in determining whether this type of oculomotor gain control occurs. Our data show that this is, indeed, the case: humans alter their eye drifts in the Snellen test in a way that enhances power in the relevant spatial frequency range.

In terms of microsaccades, we have previously observed control of microsaccades as small as 20⁰. But the distance between the optotypes in a 20/20 line is only 10⁰, and it is unclear whether humans are capable of precisely controlling saccades this small. Given that pattern vision is enhanced

around the center of gaze (Poletti et al., 2013) some way of redirecting the line of sight from one optotype to the next seems necessary, either via directional drift, larger microsaccades that skip optotypes, or smaller, sequential microsaccades. We show that the latter scenario occurs: humans tune down their microsaccades to match the spacing between optotypes.

In the revised manuscript, we have entirely rewritten this section to better explain our assumptions and the rationale for the study. The relevant text is on lines 46-65.

Fig. 1. Increase space between panels a and b, the figure looks crowded. Unclear what the significance of the vertical pink lines in panel b is | onset of microsaccades?

We have increased the space between the panels and reorganized some of the labels to ensure that the figure is not overcrowded. The shaded pink regions represent the periods of microsaccades. We have revised the caption to better explain what the various symbols and graphic conventions represent.

Line 95. Tuned "to" the task. We have rewritten the sentence.

Line 105. This paragraph is a little less intuitive. Whereas the reduction of drift speed in the Snellen task makes sense, I am not sure what the significance of curvature of drift is in general. It might be helpful to explain this before outlining results (maybe moving the logic outlined in lines 123-124 or line 153 following up?).

Ocular drift changed in two important ways in our experiments. It became slower and curved more often than during fixation. The two effects are actually intertwined, because the average speed on each axis depends on how curved is the trajectory. Intuitively, a change in curvature has two primary consequences: it implies that drift tends to change direction more often, ending up displacing the line of sight by a smaller amount than a less curved trajectory with similar speed. And it also implies that the velocity vector changes direction frequently, effectively spreading the spatial power of the optotype on each axis across a broader range of temporal frequencies on the retina.

More formally, these effects are well captured by changes in a single parameter | the diffusion constant, D | in a Brownian motion model of drift, a good approximation of ocular drift. The diffusion constant defines how rapidly the line of sight tends to move away from its current position, thus incorporating both the consequences of speed and curvature. In this model, the amplitude of the resulting luminance modulations on the retina can be estimated analytically:

$$Q(k; \omega) = \frac{2|k|^2 D}{|k|^4 D^2 + \omega^2}$$

where Q is the fraction of the power in the external stimulus at spatial frequency $k = (k_x, k_y)$ brought by eye movements to temporal frequency ω on the retina. This function peaks at $|k| = \frac{\omega}{D}$. Thus, a smaller diffusion constant, as in a more curved drift, enhances the amplitude of modulations at higher spatial frequencies.

Given the Reviewer's comment, we have revised this section to make sure that the importance of

drift curvature is clear to the reader. As suggested by the Reviewer, we qualitatively describe the impact of a more curved drift for the retinal input (last paragraph in the Introduction, lines 66-68). We then explain more clearly that the diffusion constant captures both speed and curvature and determines the characteristics of luminance modulations (lines 97-122) and refer back to this in analyzing their impact on temporal modulations (lines 123-137).

Line 118. Delete "well". This sentence (now line 101) has been changed as suggested by the Reviewer.

Fig. 4c. Can the authors explain the increase in performance at the +25' eccentricity?

This is an interesting question to which we do not have a clear answer. The majority of our subjects performs better on the last optotype to the right relative, for example, to the adjacent one (the optotype at 15° eccentricity). However, this effect is statistically significant in only 2 subjects; across the population, comparison between proportions correct at the two extreme of the Snellen line (-25° and +25°) is not significantly different.

We do not have a clear explanation for this trend in the data, but we can speculate that a number of factors are at play. One consideration is that the first and last optotypes are affected by less crowding than the others, which should facilitate identification although evidence of crowding in the fovea is somewhat conflicting (Pelli and Tillman, 2008; Levi, 2008). This effect may combine with asymmetries in cone density distributions, which appears to be higher in the nasal retina (Curcio et al., 1990, ; the positive eccentricities in our study), as well as in the allocation of attention, given the elongated rightward attentional span present in left-to-right readers (Rayner et al., 2010). We have commented on this issue on page 14 of the revised manuscript.

Fig. 4c/d. Overall, there appears to be large individual variability between subjects in terms of performance/acuity as a function of eccentricity. Were there any effects of age or other factors that might explain these differences?

To clarify: we assume the Reviewer refers to the data measured under retinal stabilization, when the eccentricity of each optotype remained constant on the retina. In the normal, unstabilized condition, the optotypes moved on the retina and performance was more or less uniform across eccentricity.

Perhaps, part of the issue here is that the overall impact of retinal stabilization (the average over eccentricity) varied across subjects. While all subjects exhibited significantly reduced performance under retinal stabilization, some were more strongly affected than others. Once these global differences are discounted, the individual changes with eccentricity were less striking, as shown in Fig. 4d, with all subjects exhibiting enhanced performance for the optotypes closest to the very center the gaze (those at 5°).

Individual differences, however, do persist: some subjects perform better in the nasal hemifield (-5°) and others in the temporal one (+5°), plus clear differences occur in performance for the last optotype (+25°). These differences do not seem to be caused by obvious factors. All our subjects were in a relatively narrow age range and possessed at least 20/20 acuity, as measured by a standard Snellen test. We also did not notice any clear sex difference. The most likely ~~conclusion is that~~

these differences may be related to individual patterns of sensitivity and/or individual capabilities in allocating attention across the foveola. We now comment on this issue on page 14 of the Results.

Discussion. I think there is a missed opportunity here in not pointing out at least the potential relation between deficits in eye movements (such as square-wave jerks, frequently found in many patients with neurological or psychiatric disorders) and acuity. Is there any indication in the literature that such a relation might exist? If not, this would certainly be an important future research direction.

We fully agree with the Reviewer that this is an interesting implication of our study and have expanded the text to discuss the possible relation between deficits in visual acuity and control. On page 19 of the Discussion, we now note that abnormal eye movements and impairments in spatial vision co-occur in various conditions, including amblyopia, congenital nystagmus and Parkinson's disease, to mention a few cases. While abnormal rotational eye movements can take many forms, including larger drifts, oscillatory movements, saccadic intrusions, and increased rates of square wave jerks, they all appear to be frequently accompanied by reduced acuity. The nature of the link between oculomotor behavior and visual functions at this scale has been under-investigated, and our results stress the need to examine this coupling in greater depth.

1.302. How was it determined that subjects were emmetropic? Given individual differences, this is important information to add.

Participants were screened with a standard Snellen eye chart test before the beginning of the experiments. They were determined to be emmetropic, if they were able to correctly report at least 6 of the 8 optotypes on the 20/20 line of a Snellen chart. As in the actual experiments, subjects viewed the stimuli monocularly with the right eye, while the left eye was patched. We updated the text on page 22 of the Methods to provide a more detailed description of the procedure.

Reply to Reviewer 2

Reviewer 2 found the "set of experiments elegant", "the execution of the study exemplary", and "the data clear". He/she agrees that "the demonstration that FEMs substantially impact acuity in the Snellen task is of potential clinical importance." The Reviewer's main concern is with the way the paper is presented, as "it may overemphasize the novelty and surprise of the findings."

We thank the Reviewer for the thoughtful reading of the manuscript. We realize that, in our effort to be concise, we ended up sacrificing the explanations of the novelty and significance of our results. In this resubmission, we have thoroughly revised the manuscript to clarify both the importance of our findings and how they relate to the previous literature. All the Reviewer's comments were incorporated in the revised text, as explained in detail below.

In brief, our study is important for two primary reasons: (1) because it reveals new types of oculomotor control; and (2) because it quantifies how much these behaviors jointly contribute to the outcome of the most common test of visual acuity, the Snellen test. This test is routinely used worldwide, but oculomotor influences are virtually never considered.

Regarding oculomotor control: The Reviewer's report focuses on microsaccades, one of the two main types of saccadic eye movements; we explain below how our claims on microsaccades add to the current knowledge in the literature. However, in terms of oculomotor control, the most surprising finding of our study is not on microsaccades, but on ocular drift, the other main type of saccadic behavior. We show that ocular drift is controlled in a way that is functional to the task. Control of drift has long been speculated (e.g., Steinman et al. (1973)) and it is known that vestibulo-ocular movements can extend to this small scale (Poletti et al., 2013). Previous studies have shown that humans are sensitive to the luminance modulations resulting from ocular drift (e.g., Rucci et al. (2007)), so active, task-dependent control of ocular drift would be beneficial. However, experimental evidence for this has remained elusive. Here we show that humans actively tune their drift in the Snellen test, enhancing the amplitude of temporal modulations on the retina in the frequency range relevant to the task. This result, as well as the active tuning of microsaccades to a scale much smaller than previously reported (point 2 below in this letter), represent important novel

Regarding the oculomotor contributions to Snellen acuity: our study builds upon the work by Rordorf and colleagues, who recently reported reduced performance with high-acuity stimuli viewed in the absence of retinal image motion (Ratnam et al., 2017). These authors examined the percentage of correct responses with an isolated, i.e., uncrowded, fixed-size optotype. They did not measure the acuity loss resulting from lack of visuomotor consequences, how much the optotype should be enlarged to maintain performance, nor how much eye movements account for acuity in a standard (crowded) Snellen chart. Furthermore, in this very interesting previous study, the stimulus was flickered at low temporal frequency bypassing the eye optics. However, earlier classical studies that used more natural stimulation reached opposite conclusions, leaving open the question of whether eye movements affect performance during normal execution of standard acuity tests. Here, we build upon this previous literature in two ways: (1) by showing that contrary

to the classical literature and extending Roorda's eye movements play an important role in standard acuity measurements, when stimuli at the limits of visual acuity are viewed normally through the eye optics; and (2) by quantifying in terms of acuity how much eye movements actually contribute to the outcome of a Snellen eye-chart test, the most common visual test.

We believe these results, both the quantification of oculomotor contributions to Snellen acuity and the finding of novel forms of control that enable these effects, represent important advances in the

1. Study by Ratnam et al, 2017. The authors write, "It has remained unclear whether FEMs enhance acuity under natural viewing conditions" (line 224). But a recent study (Ratnam et al., 2017, cited by the authors) demonstrated impaired acuity when the image was stabilized on the retina, similar to the present study. The studies cited to the contrary were from the 1950s/ 60s and are clearly out of date due to technological advances since then.

This sentence was ambiguous| it was not our intention to diminish the importance of Ratnam et al (2017), which was, in fact, also cited in the sentence immediately preceding this one. We have completely rewritten this section to remove ambiguity and explain how our work relates to the previous literature. In the new version of the manuscript, three new paragraphs replace this section (paragraphs 5-7 in the Discussion, pages 17-19).

The paragraph reviews the previous literature. We agree with the Reviewer that the studies by Cornsweet and Tulunay-Keeseey are now outdated because of the recent technological advances (Riggs et al., 1953; Tulunay-Keeseey, 1960), and we point that out in the following paragraph. However, these two papers are the only ones we know that examined the impact of retinal image motion at the limits of visual acuity with stationary external stimuli normally viewed through the eye optics. These papers are also important to cite from a historical perspective, because they led to the long-standing belief that small eye movements play little role in spatial vision.

The second paragraph focuses on significance of our findings and how they relate to the more recent literature, particularly the work by the Roorda's group. The third paragraph discusses possible mechanisms underlying these results, mentioning both the model put forward in Ratnam et al. (2017) (Reviewer's point 7, below) and the idea of temporal tuning of the visual flow onto the retina. We have also expanded the Introduction to better explain how our current results go beyond the current literature.

2. Role of microsaccades. The authors write that small microsaccades are widely believed not to be controlled (line 102) or commonly believed not to be controlled (line 246). These statements are given without references. However it is my understanding that the senior author and his colleagues have shown in several high-profile papers that microsaccades are precisely controlled.

We thank the Reviewer for pointing out this issue| the text did not make sufficiently clear that we were referring to very small microsaccades. Previous work from our laboratory has indeed shown that microsaccades tend to be precisely directed. Critically, however, this degree of control has always been observed for microsaccades in larger amplitude ranges than those present in this study,

as explained in detail below. Here we show that subjects are capable of selectively narrowing their microsaccade distributions to much smaller amplitude ranges than those reported in our previous studies, and that even the smallest microsaccades are precisely controlled. This is important because it raises the hypothesis that subjects unable to precisely control microsaccades this small may have reduced acuity.

For example, Poletti, Listorti, and Rucci (2013) have shown that microsaccades can position stimuli precisely on the retina, at distances down to 5 arcmin. And Ko, Poletti, and Rucci (2010) have shown that microsaccades are tuned to a needle threading task, and closely track the virtual thread, at a spatial scale of apparently < 10 arcmin.

In both these previous studies, microsaccades were considerably larger, with average amplitudes of 20° . In hindsight, we now see that these experiments were not suited to examine the smallest microsaccades because of the elongated nature of the stimuli used in these tasks (a horizontal bar in Ko et al, 2010; two vertical bars in Poletti et al. (2013)). These stimuli did not require subjects to execute very small microsaccades and also did not allow assessment of the precision of these movements.

In Poletti et al. (2013), exceedingly few microsaccades had amplitudes around 10° (their supplementary S3A). Note that, when the eccentricity of the stimuli was reduced, microsaccades remained larger and became less frequent (Fig. S4). In Ko et al. (2010), microsaccades did become smaller as the thread approached the needle, but the average amplitude at the end of the task was still close to 20° (Fig. 3 in Ko et al. (2010)), which matched the microsaccade amplitude measured from the same observers during sustained fixation (Fig. 2). In this study, microsaccades of only 10° or less did occasionally occur, and their frequency increased a bit as the thread approached the needle (Fig. 3b), suggesting that also these small movements were targeted. But given the configuration of the stimulus (the long horizontal thread) and that these movements were mostly horizontal, it was not possible to determine if that was the case and how precise they were.

For all these reasons, we were surprised to learn that precisely controlled saccades of only 10° play a functional role in the Snellen test. We can see how, given the previous literature, this may come across as less striking than the control of ocular drift. However, it reveals an important type of motor contribution to the outcome of the Snellen test. We have revised the text to clarify how our results build upon the previous literature. The pertinent paragraphs in the Introduction (lines 62-65) and the Discussion (lines 318-334) have been rewritten. In the Results, we have moved the section on microsaccades to follow the analysis of drift in order to give more importance to the latter. We also remind the reader of the differences of these results relative to the previous reports on page 13 in the Results section.

3. Eye movements are too coarse. The authors argue, one would not expect [the known MS enhancement of high SFs] to apply to visual acuity, as the reported characteristics of FEMs are too coarse (line 229). It would be helpful to cite the papers showing coarse FEMs, and to explain what those FEM characteristics quantitatively predict for SF enhancement. But this claim also seems at odds with the previous demonstration that FEMs enhance acuity (Ratnam et al., 2017).

We apologize for this issue. This was a critical passage in the text; it was supposed to explain the rationale for the study, but we realize that it remained very unclear. We have entirely rewritten this section in the revised manuscript.

Our point is that the characteristics of directional eye movements| both eye drift and microsaccades| typically reported by studies in the field do not seem well-suited to enhance visual acuity.

In terms of ocular drift, the motion normally observed during fixation (e.g., Cherici et al. (2012)) delivers luminance modulations to retinal receptors that peak at much lower spatial frequencies than those needed to report stimuli at the limits of visual acuity. An example is given in Figure 3D, which shows how the power of the input signal measured during sustained fixation declines rapidly in the range relevant to the Snellen test (primarily 30 cycles/deg for the 20/20 line). We knew from previous studies (Kuang et al., 2012), that a reduction in the drift's diffusion constant (i.e., a slower and more curved drift) would have the consequence of shifting power to a higher range of spatial frequencies. Given the results from Kuang et al. (2012) and those from Ratnam et al. (2017), we were interested in determining whether this type of oculomotor gain control occurs. Our data show that this is, indeed, the case: humans alter their eye drifts in the Snellen test in a way that enhances power in the relevant spatial frequency range.

In terms of microsaccades, as explained above, we have previously observed control of microsaccades as small as 20^0 . But the distance between the optotypes in a 20/20 line is only 10^0 , and it is unclear whether humans are capable of precisely controlling saccades this small. Given the selective enhancement around the preferred retinal locus that we have previously observed (Poletti et al., 2013) some way of moving the preferred locus from one optotype to the next seems necessary, either via directional drift, larger microsaccades that skip optotypes, or smaller, sequential microsaccades. We show that the latter scenario occurs: humans tune down their microsaccades to match the spacing between optotypes.

These issues are now described in the last two paragraphs of the Introduction, which explain the rationale and goals of the study. They are then addressed in deeper detail in several points of the Discussion, first in discussing the consequence of oculomotor control (second and third paragraphs) and then in describing the consequences on the retinal input (pages 15-16).

Questions:

1. Was the greater directional selectivity during the Snellen test due only to the microsaccades moving between optotypes (reading left to right across the row), or were microsaccades within optotypes also more directionally selective?

The directional bias was much weaker when we restricted our analysis to microsaccades smaller than 10^0 that maintained gaze within optotypes. In most subjects, the resulting distributions did not show any directional selectivity. On average across subjects, the microsaccade distribution was slightly asymmetric with a preference toward the right, but this bias was weak (Fig. 1a below). These results suggest that smaller microsaccades were used to explore individual optotypes. However, we could not find any clear relation between the direction of these microsaccades and the

orientation of the optotype itself. We have commented on this point on lines 204-207 of the revised manuscript.

2. Why are some statistical tests reported for group data and some for individual observers? Please explain or report both for all tests.

All effects are reported as group statistics with the probability values and statistical details given in the text. For the most important results, such as the change in diffusion constant and loss of acuity, we also report individual statistics in the text to point out that these are very robust effects that are visible in the data from each individual subject. We added a note in the Methods to clarify this (lines 531-533).

3. Line 89: Why Kuiper's test instead of K-S test? (which was used in seemingly the same analysis in line 113)

The K-S test is not circularly invariant and, therefore, cannot be directly applied to circular distributions. Kuiper's test is closely related to the K-S test and is circularly invariant. It, thus, seems a more suited statistical test for comparing the distributions of microsaccade directions measured in different conditions. We now comment on this on lines 558-561 of the Methods.

4. Line 92: How was the PRL determined?

By preferred retinal locus of fixation we mean the center of gaze as estimated by our gaze-contingent procedure. As in our previous studies, we used a gaze-contingent calibration procedure in which the estimated location of gaze was displayed in real-time, and subjects iteratively eliminated offsets using a joystick. We have previously shown that this procedure greatly reduces the dispersion of gaze localization relative to standard calibration procedures (Poletti and Rucci, 2016). Note that this procedure does not allow us to know where the PRL is located on the observer's retina, but it does determine quite accurately the location on the monitor from which this region receives visual stimulation. We have modified the text to explain the terminology and better explain the calibration procedure (lines 469-476).

5. Figure 2c: What kind of binning was done on the "Average Time" axis, or how was the visualization along that axis created?

Like in a normal eye exam, no time limits were posed on the completion of the task. As a consequence, each subject proceeded at their own pace, with some subjects taking substantially longer than others. The time axis in Fig. 3d (former Figure 2c) discounts this individual variability by normalizing each trial by its own duration. We then binned the data by subdividing the normalized time into six consecutive intervals with equal duration. The time reported on the y axis of the plot represents the average time across all data points in the corresponding bin. We have modified the text in lines 561-567, to clarify the procedure.

6. Figure 4c: Why is the "stabilized" function asymmetrical?

This is an interesting question to which we do not have a clear answer. The majority of our subjects performs better on the last optotype to the right relative, for example, to the adjacent one (the

optotype at 15° eccentricity). However, this effect is statistically significant in only 2 subjects; across the population, comparison between proportions correct at the two extremes of the Snellen line (-25° and $+25^\circ$) is not significantly different.

We do not have a clear explanation for this trend in the data, but we can speculate that a number of factors are at play. One consideration is that the first and last optotypes are affected by less crowding than the others, which should facilitate identification| although evidence of crowding in the fovea is somewhat conflicting (Pelli and Tillman, 2008; Levi, 2008). This effect may combine with asymmetries in cone density distributions, which appears to be higher in the nasal retina (Curcio et al., 1990, the positive eccentricities in our study), as well as in the allocation of attention, given the elongated rightward attentional span present in left-to-right readers (Rayner et al., 2010). We have commented on this issue on page 14 of the revised manuscript.

7. Ratnam et al. 2017 propose a model for how FEMs enhance acuity related to the way that the cone mosaic samples the drifting image. How does this model relate to the authors explanation? This should be discussed.

We have added a paragraph on page 18 of the Discussion. The two models are not mutually exclusive, but they rely on somewhat different principles: the sequence of slightly shifted retinal images for Ratnam et al. (2017) and the spatiotemporal distribution of power in the luminance flow to the retina in our proposal. The proposal by Ratnam is, perhaps, better suited than ours to explain results obtained with sequences of flashes at relatively low temporal frequencies (as in their experiments). But with naturally stationary stimuli both proposals can work and the redistribution of power we report {which is equivalent to an enhancement in contrast| is unavoidable on the retina.

8. It may not be necessary for this report, but I wondered, what are the characteristics of eye movements during stabilized viewing? Do they differ from those during normal viewing?

Eye movements under retinal stabilization were similar to those observed during normal viewing. The main noticeable difference was in the span of drift, which was larger in the stabilized condition (see Fig. 1b below), but well within the range in which it can be accurately counteracted by our stabilization apparatus. We have not included these data in the revised manuscript, but we will be glad to do so if the Reviewers think they can be useful to ~~readers~~.

Figure 1: Eye movement characteristics. (a) Directional distribution of microsaccades smaller than 10^0 that maintained gaze on an optotype. Only a weak bias to the right was present (50% reduction in the percentage of rightwards microsaccades; $p = 0.0026$; two-tailed paired t-test). (b) Eye movements under retinal stabilization. Drift was faster and less curved under retinal stabilization, as shown by the increase in diffusion constant. The resulting speed (mean 0.87 /s) was well within the range in which our apparatus guarantees accurate stabilization. In both panels, data represent mean and SEM across subjects. Triangles represent individual subjects data.

Reply to Reviewer 3

Reviewer 3 found our results to be "interesting and their implications broad", but noted that their novelty needs to be clarified. The Reviewer also commented on several aspects related to the analysis and characteristics of eye movements. This latter set of comments focuses specifically on one of our conclusions (the tuning of eye movements) without impinging on the other main contribution of the study (the quantification of oculomotor influences to acuity). In the effort to address all the points raised by the Reviewer, we have rewritten large sections of the text and introduced new experimental data. This letter explains in detail our changes.

Novel advance. "It isn't clear the extent to which [these results] represent a major advance on earlier published work. The authors give a sense of such an advance (e.g. by suggesting FEMs were previously generally considered too "coarse"), but it is left relatively vague and with un-cited assertions about the prevailing view in the field. This group has previously shown that FEMs have important effects on sensitivity to fine detail through their effects on the spatiotemporal content of incoming images. The novel advance here could thus perhaps be clarified."

We realize that the manuscript should have done a better job in explaining the significance and novelty of the results. In this resubmission, we have thoroughly revised the text to ensure that the novel advances as well as the rationale for the study are clear. In brief, there are two major novel advances introduced by this study: (a) the observation of highly specific tuning in eye movements, both ocular drift and microsaccades; and (b) the assessment of oculomotor contributions to visual acuity.

Regarding the first point: we report that, when confronted with stimuli at the limits of visual acuity, humans (a) actively tune their ocular drift, and (b) perform targeted microsaccades at a scale much smaller than previously observed (see below in this letter for a detailed explanation of the microsaccade). The results on drift are particularly striking as they provide the first demonstration that ocular drift is controlled in a way that is functional to the task. Active task-dependent control of eye drift has long been speculated (e.g., Steinman et al., 1973) but experimental evidence has remained elusive. Previous studies with stimuli in intermediate spatial frequency ranges| far from resolution limits (e.g., Rucci et al., 2007; Boi et al., 2017)| have shown that humans are sensitive to the luminance modulations resulting from ocular drift, but these modulations normally peak at much lower spatial frequencies than those needed to report stimuli at the limits of visual acuity (Fig. 2d). Here we show that observers actively control their drift in the Snellen test, enhancing the amplitude of temporal modulations on the retina in the frequency range relevant to the task.

Regarding the second point: Our results indicate that high visual acuity is not a purely visual accomplishment but the outcome of a visuomotor process that requires active motor control. This observation redefines what the Snellen test actually measures. The Snellen eye chart is arguably the most common visual test; it is routinely used worldwide both in clinical and scientific settings. The outcome of this test is traditionally regarded as an evaluation of the quality of the eye optics,

and oculomotor in uences are virtually never considered. Our study shows that normally tuned eye movements play a critical role: they are responsible for at least 0.15 logMAR, two lines of the Snellen chart.

Both set of results represent important novel contributions. In addition to advancing current knowledge on the control processes and visual functions of eye movements, they also have important clinical implications, as they suggest that suboptimal control of eye movements may lead to reduced acuity. In this revision, we have modi ed extensively the manuscript to highlight the most important messages, explain the signi cance of our results, and clarify how they advance the literature. The last three paragraphs in the Introduction now provide the background and describe the rationale of the study. The eight paragraphs in the Discussion relate our results to the literature and explain their novelty.

Tuning of eye movements. The Reviewer made several comments on the characteristics of eye movements and how we reached the conclusion that they are tuned. Most of these comments focus primarily on microsaccades and are addressed in detail below. In addition to the changes listed here, we have also moved the results on drift to precede those on microsaccades. This was done to ensure that the order of presentation of the materials does not take the reader's attention away from the related to ocular drift, which we think provide the most surprising aspects of oculomotor tuning.

\In the half of the paper, FEMs are compared between the Snellen and conditions. Many di erences are observed, and they are interpreted as evidence for "tuned" control of FEMs due to the need for high acuity in the task. These conditions, however, are di erent in many important ways, not just in the need for high acuity. The comparison seems to entirely be one of apples and oranges. Some di erences and observations are as follows: "

Sustained on a marker is the condition used by most studies on eye movements. In looking for a reference to use as comparison, it makes sense to start from the most common condition. However, we do agree with the Reviewer that adding other baselines would strengthen our claims and have now included new data sets, as explained below. We have also revised the text to emphasize that multiple observations (e.g., the microsaccade peak amplitude matching the optotypes spacing, the landing distance of microsaccades, etc.) stand on their own and do not depend on comparisons with external references.

a) Although not stated in the main text, the " xation task" turns out not to be a separate, comparable task (e.g. one that di ers only in the need for acuity,) but is rather the initial period of in the Snellen task, before the array of optotypes is presented. At a minimum, this should be made clear early in the main text, but it could also impact on the oculomotor behaviour observed:

| What if subjects consider that epoch to be less important, as they anticipate the "real" task, and maintain less precise control? (the details of how was enforced were also not provided).

The Reviewer raises an interesting point. The experiments were designed in this way (with preceding Snellen) because all our previous separate attempts to identify anticipatory task influences on ocular drift have always given negative results. However, we agree with the Reviewer that, in principle, the characteristics of eye movements during the initial fixation period could be affected by anticipation of the subsequent Snellen task. If present, such effects, may act as confounds.

Given the Reviewer's concerns, our resubmission contains three new sets of experimental data. First, we now report the results of a separate, stand-alone control experiment in which subjects were explicitly asked to maintain very accurate fixation. In this condition, fixation was maintained for the entire duration of the trial and was not followed by the Snellen task. As shown in Supplementary Figure S1, each individual observer continued to exhibit significantly larger eye drifts and microsaccades relative to the Snellen test, even though, paradoxically, all they were asked to do was to maintain steady gaze on a dot.

Second, we have added baseline comparisons from two larger pools of subjects, who, in separate dedicated experiments, either maintained fixation on a dot for the entire trial duration (no other task) or freely examined images of natural scenes. These subjects did not undergo the Snellen test and within-subject statistics was not possible. But comparisons between subject populations confirm tuning of both drifts and microsaccades in the Snellen test.

Third, we have included results of a control experiment in which we compared eye movements when subjects performed two different tasks with the 20/20 line: (a) the standard Snellen task in which subjects reported the orientation of each optotype, as in our main experiment; and (b) a judgment on the orientation of the entire line, whether the line was tilted by 4° clockwise or counter-clockwise. In this experiment, the stimuli presented in the two conditions were identical, but only the judgment task required high visual acuity. In keeping with the other results, we show that every individual subject exhibited smaller microsaccades and drifts in the Snellen task (Supplementary Figure S2). These data further support our conclusion that humans tune their eye movements to reach high visual acuity.

The manuscript has been modified in multiple points to incorporate these new data. The most important changes are on pages 9-12 of the Results where the new results are reported and on pages 25-26 of the Methods, where the control experiments are described. Two new supplementary Figures have been included (Figs. S1 and S2). We have also revised the manuscript to add details about the fixation task (Methods, page 25) and to state upfront that the Fixation condition data in Figs. 2 and 3 refers to data collected during the initial portion of the each trial (lines 83-85).

Indeed, although the distribution of microsaccade amplitudes in the Snellen task was shifted toward smaller eye movements compared to fixation in *this* data set, it looks very similar to those seen during fixation in other studies (e.g. compare with Box 1 in "Microsaccades: a neurophysiological analysis" by Martinez-Conde et al., or "Microsaccades during reading" by Bowers and Poletti). Similarly, stimulus and task-related asymmetries in microsaccade directions are also well established. The Rucci lab is certainly a world-leader in this area, and thus I expect that there are important advances in the result here to be appreciated, but this needs to be made clearer to the

reader."

It was not our intention to come across as implying that microsaccades this small have never been observed. Microsaccades vary considerably in amplitude across observers, and classical studies reported extremely small amplitudes, as remarked by the Reviewer. Our main point here is not about the existence of such small saccades, but the finding that they are precisely directed to an extent not previously reported. The microsaccade amplitude distribution in Fig. 3 is shown primarily to observe that microsaccades in the Snellen task match the spacing between optotypes. The more important observation comes later, when we show that these small microsaccades shift gaze from one optotype to the next. We have revised the text to avoid possible ambiguity and added references to other review articles on microsaccades. The relevant material is on the second and third paragraphs in lines 318-334.

b) The Snellen array is arranged horizontally, and subjects were instructed to report the identity of the optotypes one-by-one with a joystick and from left-to-right (this important detail was also not stated until the methods). This makes sense| it's consistent with the clinical version of the task| but it is probably not surprising that in such a task the FEMs showed a left-to-right progression. That is, again, it might have nothing to do with the need for acuity per se. In comparison, the task is radially symmetric. Perhaps the same pattern of FEMs would occur with any task requiring observers to shift their average gaze position rightward over time (as was the case in the paper on reading cited above).

We are not sure we follow the Reviewer's comment, perhaps there has been a been a misunderstanding here.¹ We fully agree with the Reviewer that several factors| some unrelated to acuity| could contribute to the left-to-right progression. But this is marginal to our claims; we are certainly not arguing that we understand all the factors responsible for this sequence. Our main observations are that, irrespective of its causes, this behavior is: (a) functionally relevant; and (b) mediated by targeted microsaccades at a smaller scale than previously reported (see reply to the following point, below).

The functional importance of the microsaccade sequence is highlighted by the acuity impairments observed when the sequence is disrupted, as under retinal stabilization, when the motion of the stimulus counteracts the visual consequences of eye movements on the retina. The larger impairment at larger foveal eccentricities (Fig. 4d) likely follows from the impossibility to recenter gaze via microsaccades.

To further corroborate the importance of this behavior, in this resubmission we have added the results of another control experiment in which subject were asked to maintain gaze at the center of the monitor during execution of the Snellen test. These new data confirm that performance drops considerably when subjects do not implement their normal microsaccade strategy. These results are important because they suggest possible acuity impairments in observers who are not capable of executing sequences of microsaccades this small. The new data are summarized in Supplementary

⁰ To eliminate one possible point of confusion: in the Bowers and Poletti article mentioned by the Reviewer, microsaccades were primarily leftwards, not rightwards.

Figure S3. We have also revised the text on page 14 to eliminate possible ambiguity and emphasize the relevant aspects of this behavior, particularly its consequences.

c) As an indication of how targeted microsaccades were, the authors state that "when the oculomotor traces recorded in the Snellen test were randomly replaced by traces of equal duration acquired during the average distance between each optotype and the PRL increased drastically (Fig. 2d)." Do they mean the average distance between the PRL and the *nearest* optotype? I presume so, because otherwise I would expect the opposite result.

We have revised the section to include the analysis suggested by the Reviewer and better explain our results. The new Fig. 3e (which replaces former Fig. 2d) reports the landing distance relative to the closest optotype averaged across all microsaccades. This was achieved by determining, for every microsaccade, the distance to the closest optotype, and then averaging this distance across all microsaccades. Data points in the figure represent averages of these mean values across subjects. The figure compares the mean landing distance measured in the Snellen task to those obtained when microsaccades were (a) substituted by microsaccades randomly selected from the pool recorded during the Snellen test and (b) randomly permuted across all the Snellen microsaccades. In both cases, the new microsaccade was positioned so to possess the same starting location as the original one.

The revised text (but not the figure) also reports the results of analyses in which, for every optotype, we estimated the closest microsaccade landing and then averaged across all optotypes. This analysis focuses on the most effective microsaccade for each optotype, whereas the one described above quantifies how targeted microsaccades are on average. They give similar outcomes: in both cases, distances increase when the Snellen microsaccades are randomly replaced by those executed during the Snellen test or when the Snellen microsaccades are randomly permuted. These results indicate that microsaccades are effective in bringing gaze close to the optotypes. The relevant text is in lines 190-204 of the revised manuscript.

d) Related to these targeted microsaccades. How does this differ from previous reports of targeted microsaccades (e.g., correcting for drift during the Snellen test)? The suggestion is that those observed here are much smaller, but as stated above, the advance is asserted to be true without sufficient quantitative comparison to earlier work. For example, as I have noted, the distributions of microsaccade amplitudes do not seem much smaller than in earlier work."

To avoid possible confusion: the question here is not whether microsaccades of 10^0 occur, but their landing accuracy relative to the target. We are not aware of any previous study reporting precisely targeted microsaccades at this scale. Examination of microsaccade accuracy requires localization of the line of sight on the stimulus. Because of the uncertainty inherent in standard methods for gaze localization (typically around 1°), most previous studies from other laboratories did not possess the resolution necessary for examining the precision of saccades this small.

Our results rely on gaze-contingent calibration techniques specifically designed, over the course of a decade, to enable accurate localization of the line of sight. These methods have been shown to

reduce uncertainty in gaze localization by approximately one order of magnitude (Ko et al., 2016). Using similar methods, two previous studies have observed precisely targeted microsaccades (Ko et al., 2010; Poletti et al., 2013). However, in both these studies, the tasks did not require gaze shifts as small as the ones observed in the Snellen test, and microsaccades were considerably larger, with average amplitudes of 20° . In Poletti et al. (2013), exceedingly few microsaccades had amplitudes around 10° (Fig. S3A in Poletti et al., 2013). In Ko et al. (2010), microsaccades became smaller as the thread approached the needle, but the average amplitude at the end of the task was still close to 20° (Fig. 3 in Ko et al., 2010), which matched the microsaccade amplitude measured from the same observers during sustained fixation (Fig. 2 in Ko et al., 2010). Given this previous literature, we were surprised to learn that (a) the Snellen test would elicit frequent microsaccades of only 10° ; (b) that these microsaccades are precisely directed; and (c) that they play a functional role in the test outcome.

We have modified the manuscript to clarify how our results build upon this previous literature. The relevant material is included on pages 11-12 in Results and 16-17 in Discussion of the revised manuscript. Given the Reviewer's comment on the role of microsaccades in compensating for drifts during fixation we have also added a reference to the interesting study by Engbert and Kliegl (2004), which investigated this issue. This study, however, examined relative changes in gaze position, not microsaccade accuracy, and made no claim about the accuracy of microsaccades of small amplitudes.

Minor Points

Fig 4b. The x-axis is labelled "Acuity loss", but perhaps it would be clearer to label this as "optotype size?"

We have changed the label of the axis as suggested by the Reviewer.

Methods state that blinks were removed, but no details provided.

We have added more details in the Methods (page 27). The DPI eyetracker automatically labels as blinks the periods in which the system is in tracking mode and the two Purkinje images suddenly disappear.

Fig 4 caption: "red circles". There is only one red circle data point.

We have corrected the typo.

References

- Boi, M. and Poletti, M., Victor, J. D., and Rucci, M. (2017). Consequences of the Oculomotor Cycle for the Dynamics of Perception. *Curr. Biol.*, pages 1{10.
- Cherici, C., Kuang, X., Poletti, M., and Rucci, M. (2012). Precision of sustained in trained and untrained observers. *J. Vis.*, 12(6):1{16.
- Curcio, C. A., Sloan, K. R., Kalina, R. E., and Hendrickson, A. E. (1990). Human photoreceptor topography. *J. Comp. Neurol.*, 292(4):497{523.
- Engbert, R. and Kliegl, R. (2004). Microsaccades keep the eyes' balance during Psychol. Sci., 15:431{436.
- Ko, H.-K., Poletti, M., and Rucci, M. (2010). Microsaccades relocate gaze in a high visual acuity task. *Nat. Neurosci.*, 13(12):1549{1553.
- Ko, H.-K., Snodderly, D. M., and Poletti, M. (2016). Eye movements between saccades: Measuring ocular drift and tremor. *Vision Res.*, 122:93{104.
- Kuang, X., Poletti, M., Victor, J. D., and Rucci, M. (2012). Temporal encoding of spatial information during active visual Curr. Biol., 20(6):510{514.
- Levi, D. M. (2008). Crowding - an essential bottleneck for object recognition: A mini-review. *Vision Res.*, 48(5):635{654.
- Pelli, D. G. and Tillman, K. A. (2008). The uncrowded window of object recognition. *Nat. Neurosci.*, 11(10):1129{1135.
- Poletti, M., Listorti, C., and Rucci, M. (2013). Microscopic eye movements compensate for nonhomogeneous vision within the fovea. *Curr. Biol.*, 23(17):1691{1695.
- Poletti, M. and Rucci, M. (2016). A compact guide to the study of microsaccades: Challenges and functions. *Vision Res.*, 118:83{97.
- Ratnam, K., Domdei, N., Harmening, W. M., and Roorda, A. (2017). Benefits of retinal image motion at the limits of spatial vision. *J. Vis.*, 17(1):30.
- Rayner, K., Slattery, T. J., and Belanger, N. N. (2010). Eye movements, the perceptual span, and reading speed. *Psychon. Bull. Rev.*, 17(6):834{839.
- Riggs, L. A., Ratli, F., Cornsweet, J. C., and Cornsweet, T. N. (1953). The disappearance of steadily visual test objects. *J. Opt. Soc. Am.*, 43(6):495{501.
- Rucci, M., Iovin, R., Poletti, M., and Santini, F. (2007). Miniature eye movements enhance spatial detail. *Nature*, 447(7146):852{855.

Steinman, R. M., Haddad, G. M., Skavenski, A. A., and Wyman, D. (1973). Miniature eye movement. *Science*, 181(102):810{819.

Tulunay-Keesey, U. (1960). Effects of involuntary eye movements on visual acuity. *J. Opt. Soc. Am.*, 50:769{774.

Reviewers' Comments:

Reviewer #1:

Remarks to the Author:

The authors have addressed all my comments and have improved the manuscript in many other ways. I particularly appreciate the rephrasing / addition of the research questions in intro 1.68 following, which will help the reader navigate through the ms, and the additional sections added re. the drift results. I only have a short list of minor comments that I think they could address to further improve clarity.

1. Intro, 1.23, introduces the term "foveola", but in the following paragraphs (e.g., 1.30, 1.39, etc.), the authors talk about the fovea. These regions are of course overlapping but I wonder if the authors could be more consistent to clarify whether they are talking about the fovea or the central part of the fovea throughout.
2. 1.48: is "eye drift" used here in the exact same context as "ocular drift" as introduced in 1.40?
3. 1.52: no benefits of (not from) eye movements on acuity – measured how? Since the present study is the first using Snellen it is important to specify how acuity was measured in previous studies.
4. Results, 1.90: replace "motion" by "movement"
5. 1.113: delete comma after "that"
6. 1.167-69: the new part is repetitive; the following sentence already mentions that microsaccade amplitudes were half the size in Snellen test vs. dot fixation
7. 1.177: rephrase "similar differences"
8. 1.272-283: considerable emphasis is placed on an effect that reached significance in only $n=2$ (and no information is provided on effect sizes, thus it is difficult to evaluate how important this finding is); my suggestion would be to leave this paragraph out

Reviewer #2:

Remarks to the Author:

The authors have done a thorough revision and addressed all my concerns. Importantly, they have clarified the novelty and significance of their finding that ocular drift changes its spatiotemporal characteristics during the Snellen task compared to fixation. They have also better discussed the relation of their study to previous literature, in particular Ratnam 2017. And they have improved explanations of motivation, methods, etc. throughout the manuscript.

Reviewer #3:

Remarks to the Author:

The revised manuscript is improved greatly and addresses most of the key points I raised in my previous review. The additional control experiments strengthen the paper, and the improved narrative does an excellent job of delineating the novel aspects of this study compared with earlier work. There is just one important result that I could not make sense of, and would like clarified:

Major

The analysis of drift eye movements in the Snellen task shows that they provide an increase of power in the high frequencies of " $\sim 50\%$ ". The authors suggest that this reflects a task-dependent strategy to improve acuity. Confusingly, however, they then note that: "The resulting power was significantly higher than the amount that would have been given the eye movements measured in separate subject

populations, which either maintained strict fixation for the entire duration of a trial (an average 22% increase; $p = 0.002$) or freely examined natural scenes (48% increase; $p < 10^{-7}$).

So the magnitude of the effect in the Snellen acuity task (~50%) is just 2% more than that observed while freely viewing natural images (48%)? Perhaps I am misunderstanding something, but it seems to me that even if this small difference is statistically significant, the results seem inconsistent with the idea that the drift is altered to improve acuity (unless we consider "freely viewing" to be an acuity task). Can the authors clarify these two effects and re-state their view on the functional role of drift control in the Snellen task and others.

Minor points:

- Why is the 90th percentile used in some plots, rather than a median or mean? It's not a problem, but the reason for this choice could be clarified.
- The narrative speaks of different "populations" of subjects in various places. This should be "samples".

Signed,
Adam P. Morris

Reply to Reviewer 1

1. Intro, l.23, introduces the term "foveola", but in the following paragraphs (e.g., l.30, l.39, etc.), the authors talk about the fovea. These regions are of course overlapping but I wonder if the authors could be more consistent to clarify whether they are talking about the fovea or the central part of the fovea throughout.

We have changed the text to use the term "foveola" rather than the broader "fovea" whenever pertinent.

2. l.48: is "eye drift" used here in the exact same context as "ocular drift" as introduced in l.40?

It is. To avoid possible confusion, we now introduce both terms when referring to the intersaccadic wandering of the eye in line 40.

3. l.52: no benefits of (not from) eye movements on acuity measured how? Since the present study is the only one using Snellen it is important to specify how acuity was measured in previous studies.

These studies used either detection of Vernier lines or a Vernier acuity test to measure visual resolution thresholds (Riggs et al., 1953; Tulunay-Keesey, 1960). We have modified the sentence to provide further details.

4. Results, l.90: replace "motion" by "movement".

We have modified the sentence as suggested by the reviewer (now line 91).

5. l.113: delete comma after "that".

Done (now line 115).

6. l.167-69: the new part is repetitive; the following sentence already mentions that microsaccade amplitudes were half the size in Snellen test vs. dot pattern.

We have revised the text to be less repetitive. The new text is on lines 172-176.

7. l.177: rephrase "similar differences".

Done, thanks. The new sentence is on line 181.

8. l.272-283: considerable emphasis is placed on an effect that reached significance in only n=2 (and no information is provided on effect sizes, thus it is difficult to evaluate how important this is); my suggestion would be to leave this paragraph out.

We agree with the Reviewer that this paragraph gives too much emphasis to marginal considerations. We have removed the paragraph as suggested and added a note to the previous paragraph (lines 274-277).

Reply to Reviewer 3

Major point. The analysis of drift eye movements in the Snellen task shows that they provide an increase of power in the high frequencies of $\sim 50\%$. The authors suggest that this reflects a task-dependent strategy to improve acuity. Confusingly, however, they then note that: "The resulting power was significantly higher than the amount that would have been given the eye movements measured in separate subject populations, which either maintained strict fixation for the entire duration of a trial (an average 22% increase; $p = 0.002$) or freely examined natural scenes (48% increase; $p < 10^{-7}$)."

So the magnitude of the effect in the Snellen acuity task ($\sim 50\%$) is just 2% more than that observed while freely viewing natural images (48%)? Perhaps I am misunderstanding something, but it seems to me that even if this small difference is statistically significant, the results seem inconsistent with the idea that the drift is altered to improve acuity (unless we consider "freely viewing" to be an acuity task). Can the authors clarify these two effects and re-state their view on the functional role of drift control in the Snellen task and others.

There has been a misunderstanding here. Our comparisons are always relative to the eye movements in the Snellen test. That is the power of the stimulus on the retina in our sample of observers performing the Snellen task is 48% higher than the power that would have been given by the ocular drifts recorded in a separate group of subjects who freely observed natural scenes. The reason why this power increases by similar amounts ($\sim 50\%$ and 48%) when compared to fixation and free-viewing is because the ocular drifts measured in our subjects when they maintained fixation resemble those measured in the other group of subjects when they freely examined natural scenes. We have revised this section to avoid possible misunderstandings (lines 137-141).

Minor points. Why is the 90th percentile used in some plots, rather than a median or mean? It's not a problem, but the reason for this choice could be clarified.

We use the 90th percentile when referring to microsaccade amplitudes because it better captures the asymptotic limit of the distribution, which is sometimes used to define an upper threshold on microsaccade magnitudes (e.g., Kowler et al, 2011; Martinez-Conde et al, 2013). We have added an explanation in the Methods (lines 555-556).

The narrative speaks of different "populations" of subjects in various places. This should be "samples".

We changed the text as suggested by the reviewer.

References

- Riggs, L. A., Ratliff, F., Cornsweet, J. C., and Cornsweet, T. N. (1953). The disappearance of steadily moving visual test objects. *J. Opt. Soc. Am.*, 43(6):495-501.
- Tulunay-Keesey, U. (1960). Effects of involuntary eye movements on visual acuity. *J. Opt. Soc. Am.*, 50:769-774.

Reviewers' Comments:

Reviewer #3:

Remarks to the Author:

The authors have addressed my concerns. There was indeed a misunderstanding, and I believe that the revised text more clearly describes the comparison being made.

Signed,
Adam P Morris